# ADAM CAN MITIGATE CLASS IMBALANCE WITHOUT ELEMENT-WISE GRADIENT NORMALIZATION

## ABSTRACT

Adam has remained a dominant optimization algorithm in deep learning for a decade. Recent studies reveal that Adam mitigates the class imbalance by normalizing element-level gradients to balance gradients across classes. However, this interpretation relies on an assumption that gradients between different classes are fully orthogonal. In this paper, we further investigate the assumption. We observe that inter-class gradient orthogonality can be low, particularly during the initial training stages, yet Adam still mitigates class imbalance. This suggests that Adam may not reduce class imbalance by normalizing element-level gradients. Through the ablation of Adam, we further support that class imbalance can be alleviated without element-wise gradient normalization. This work reveals that, even with inter-class gradient coupling, Adam mitigates class imbalance by normalizing gradients across iterations. During early training, the model primarily fits high-frequency class data; as the loss for these diminishes, it adapts to low-frequency classes. Due to the inter-iteration normalization, the gradient magnitudes for low-frequency classes then approximate the initial high-frequency gradients. This mechanism helps Adam mitigate class imbalance. Consequently, we demonstrate that this mechanism necessitates at least layer-wise gradient normalization across iterations, since most neural networks exhibit layer-level inconsistencies between forward and backward propagation. Finally, we further explore potential limitations in Adam's ability to address the inconsistencies.

## 1 INTRODUCTION

The Adam optimizer (Kingma & Ba, 2014) is presented as an integration of the adaptive optimization algorithm with the momentum technique. The adaptive optimization algorithms (Duchi et al., 2011; Tieleman et al., 2012) perform gradient normalization to balance gradient magnitudes by scaling down large gradients and scaling up small gradients. The momentum techniques (Sutskever et al., 2013; Nesterov, 2013; 1983) stabilize noisy gradients through weight updates aligned with historical optimization trajectories. However, despite the subsequent emergence of novel optimization algorithms (Liu et al., 2019; Zhang et al., 2019; Anil et al., 2020; Chen et al., 2023; Xie et al., 2024; Jordan et al., 2024), Adam remains dominant, as it is hard to validate that these new optimization algorithms surpass Adam.

The interpretability of Adam's mechanisms can pave the way for new optimization algorithms that can perform better than it, particularly when empirical insights struggle to devise optimization algorithms that surpass Adam across massive task and model scenarios. Therefore, many researchers focus on demystifying Adam's mechanisms. Although some of them have made significant strides Zhang et al. (2020); Jelassi & Li (2022); Francazi et al. (2023); Kunstner et al. (2024), there is no work that explicitly further explains how Adam adapts to class imbalance.

In class-imbalanced learning scenarios, gradients from high-frequency classes exhibit larger magnitudes than those from low-frequency classes, which leads to slow convergence and poor performance when training a model by SGD (Francazi et al., 2023) as well as SGD with the momentum (Kunstner et al., 2024). (Francazi et al., 2023), both theoretically and empirically, demonstrates that the primary issue underlying class imbalance is the gradient imbalance across classes, and introduces per-class gradient normalization to mitigate class imbalance. However, the per-class gradient nor-

malization is almost unusable in practical tasks, since calculating gradients separately for each class significantly increases both computational and memory costs.

Interestingly, (Kunstner et al., 2024) demonstrates that Adam can effectively handle class-imbalanced data, since the element-wise gradient normalization in Adam may approximate per-class normalization by normalizing per-weight gradients. However, the holding of this approximation relies upon the assumption that the inter-class gradients are fully orthogonal: only the gradients from a single class are non-zero for each weight. Through analysis of gradient orthogonality in CNN-based and Transformer-based models, we establish two key findings: 1) Models initialized randomly usually exhibit low gradient orthogonality between classes. 2) CNNs demonstrate stronger inter-class gradient orthogonality compared to Transformers. Moreover, Adam consistently outperforms SGD on class-imbalanced data even under low inter-class gradient orthogonality. These observations challenge the idea that element-wise gradient balancing is the primary mechanism for mitigating class imbalance in Adam.

In this paper, we demonstrate that Adam can mitigate class imbalance by balancing the magnitudes of gradients across iterations. Furthermore, as gradient balancing across iterations may operate at element-level, layer-level, or coarser granularity, we demonstrate that layer-level gradients are the coarsest feasible granularity for this operation. This is because most neural networks exhibit a layer-level inconsistency between forward and backward propagation. The inconsistency can result in a gap between model properties changing and gradient-based updating. Adam may not fully address this issue. Based on the findings, we also propose introducing a scaling factor to adjust layer-level optimization dynamics, thereby addressing the inconsistency more effectively than before.

The contributions in this work can be summarized as

- This work reveals that Adam can mitigate class imbalance by balancing gradients across iterations. Although element-wise dynamics normalization may address class imbalance, its effectiveness depends on gradient orthogonality, which in turn is influenced by multiple factors such as tasks and model architecture.

- This work argues that the layer-wise dynamics normalization can address a layer-level inconsistency between forward propagation and back-propagation while Adam may not fully address this issue. The inconsistency stems from operations that have a normalizing effect, such as normalization layers, attention layers, and the softmax function, which are widely used in neural networks. A scaling factor proportional to the initialized weight magnitudes can be introduced to further mitigate this issue.

- Replacing element-wise dynamics normalization with layer-wise normalization in Adam can reduce memory consumption equivalent to the whole gradient storage of a model while coping with class-imbalanced data well.

## 2 RELATED WORK

**Model Optimization Algorithm.** The classical optimization algorithm is Stochastic Gradient Descent (SGD). SGD scales the gradients by the step size to obtain the updated values, resulting in noisy optimization dynamics that stem from mini-batches. To denoise noisy dynamics, momentum is intuitively introduced to smooth optimization dynamics for several successive iterations (Sutskever et al., 2013). RMSProp (Tieleman et al., 2012) introduces the Exponential Moving Average (EMA) to evaluate the magnitude of the average gradient as a normalization factor, avoiding the numerical explosion of the scaling factors. Adam (Kingma & Ba, 2014), as a prevalent optimizer, combines the ideas of momentum and RMSProp, optimizing the model by the consistency of the gradient direction of weights based on momentum estimation. The previous works (You et al., 2017; 2019) empirically identify layer-wise optimization imbalance during model training with a large batch size, propose the LARS and LAMB optimizers to mitigate this issue. Lion Optimizer (Chen et al., 2023) fixes the magnitude of the dynamic as 1, reducing the computational burden and further speeding up the training of large models. Adan (Xie et al., 2024) develops the Nesterov momentum (Nesterov, 2013), yielding better results for different model structures at a high memory cost. As for second-order optimization algorithms, they theoretically require no hyperparameters (e.g., learning rate) and offer more stable optimization than the first-order ones. However, most of them are difficult to parallelize and require significant computational and memory costs, resulting in

low computational efficiency. K-FAC Martens & Grosse (2015) employs a preconditioning scheme that approximates the Fisher-information matrix of a generative model represented by a neural network. Anil et al. (2020) improves the shampoo algorithm to relieve the hardware burden, while the shampoo algorithm is introduced into machine learning by Gupta et al. (2018). However, the limited mechanistic interpretability of the difference between various models hinders the evolution of efficient and adaptive optimizers.

**Optimization-Related mechanism interpretability.** Kunstner et al. (2024) suggests that Adam outperforms SGD due to better behavior on class-imbalanced data, but the corresponding mechanism interpretability of optimizers may be limited. Francazi et al. (2023) elucidate the significant negative impact of data imbalance on learning, showing that the learning curves for minority and majority classes follow sub-optimal trajectories when training with a gradient-based optimizer. The work (Jelassi & Li, 2022) illustrates that momentum can help models capture features in severely noisy data, thanks to its historical gradients, which are similar to those in the clean data. Complementing this, Zhang et al. (2020) theoretically establish that gradient normalization in Adam can converge arbitrarily faster than gradient descent with fixed step-size. Both the works (Park & Kim, 2022; Bai et al., 2022) reveal that Transformers act like low-pass filters while CNNs act like high-pass filters, which may be related to differences in gradient orthogonality across different models. The work (Park & Kim, 2022) further shows that the loss function of ViTs is more rugged than that of CNNs by comparing the Hessian eigenvalues. This suggests that Transformer-based models may be more challenging to optimize. Zhao et al. investigate the performance and stability to hyperparameters of several optimizers, suggesting that the used adaptive optimizers perform comparably.

## 3  PRELIMINARY

**Optimization Dynamics.** Optimization dynamics refer to the power to update the weights of neural networks. For example, given a model with the weights $\theta$ optimized by SGD, then the update process of a weight at $t^{th}$ iteration can be formulated as

$$\theta_{n,t+1}^{(l)} = \theta_{n,t}^{(l)} + Dynamics_{n,t}^{(l)} = \theta_{n,t}^{(l)} - lr \cdot \nabla\theta_{n,t}^{(l)}, \tag{1}$$

where the $Dynamics_{n,t}^{(l)}$ denotes the optimization dynamics for the weights in the layer $l$ at $t^{th}$ iteration, and the $\nabla\theta_{n,t}^{(l)}$ represents the corresponding weight gradients. For clarity, although $Dynamics_{t}^{(l)}$ and $\theta_{t}^{(l)}$ are tensors rather than vectors, we denote individual elements of the tensors as $Dynamics_{n,t}^{(l)}$ and $\theta_{n,t}^{(l)}$, since our analysis is irrelevant to the dimension in which the elements lie. By default, formulas are defined as element-wise operations, meaning calculations based on scalars.

**Dynamics Normalization.** Dynamics normalization refers to applying a specific normalization to an optimization dynamics to obtain a new optimization dynamics that meets particular requirements. The update process with new optimization dynamics can be represented by

$$\theta_{n,t+1} = \theta_{n,t} + Dynamics'_{n,t} = \theta_{n,t} + lr \cdot Norm(Dynamics_{n,t}). \tag{2}$$

**Orthogonal Gradients Across Classes.** This refers to the scenario where, given a model weight, only gradients from a single class will be non-zero. For instance, in a binary classification task, the dot product of the gradients for class 1 and the gradients for class 2 is zero.

**Update of Adam.** Given a model trained by Adam (Kingma & Ba, 2014) optimizer, then the update process can be formulated as

$$\theta_{n,t+1}^{(l)} = \theta_{n,t}^{(l)} - lr \cdot \frac{m_{n,t}^{(l)}/(1-\beta_2^t)}{\sqrt{v_{n,t}^{(l)}/(1-\beta_1^t)}}, \tag{3}$$

where the $v_{n,t}^{(l)}$ and the momentum $m_{n,t}^{(l)}$ can be written as

$$m_{n,t}^{(l)} = \beta_2 \cdot m_{n,t-1}^{(l)} + (1-\beta_2) \cdot \nabla\theta_{n,t}^{(l)}, \quad m_{n,0}^{(l)} = 0 \tag{4}$$

and

$$v_{n,t}^{(l)} = \beta_1 \cdot v_{n,t-1}^{(l)} + (1-\beta_1) \cdot (\nabla\theta_{n,t}^{(l)})^2, \quad v_{n,0}^{(l)} = 0, \tag{5}$$

respectively. The hyperparameter $\beta_1$ is used to tune the smoothness of the momentum $m_{n,t}^{(l)}$ while the hyperparameter $\beta_2$ is used to tune the smoothness of the momentum $v_{n,t}^{(l)}$.

# 4 METHODOLOGY

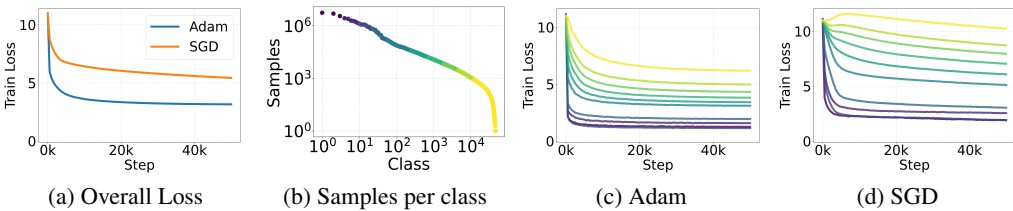

(a) Overall Loss     (b) Samples per class     (c) Adam     (d) SGD

Figure 1: Comparison of Adam vs. SGD when training GPT-2 on class-imbalanced data (i.e., WikiText-103). **(a)**: Overall training loss: Adam vs. SGD. **(b)**: Distribution of sample sorted by class frequency. We divided the data into 10 groups in order, with each group accounting for approximately 10% of the data. The groups are labelled with colors. **(c)** and **(d)**: Training loss of 10 groups: Adam vs. SGD. The results suggest that Adam outperforms SGD on class-imbalanced data.

In this section, 1) We first illustrate the results of different models trained with Adam versus SGD (Fig. 1), demonstrating Adam's effectiveness in handling class-imbalanced data. Then, 2) we investigate Adam's mechanism for addressing class imbalance, revealing that it can mitigate this issue by balancing layer-level optimization dynamics across iterations. Finally, 3) we analyze the role of layer-wise dynamics normalization in models and propose improvements to this mechanism.

## 4.1 THE ROLE OF ADAM IN CLASS IMBALANCE SCENARIOS

The previous work Kunstner et al. (2024) demonstrates that Adam can mitigate class imbalance, which is a key advantage of Adam. For example, language data are imbalanced as some words are much more frequent than others, while vision data are always imbalanced in the real world. As shown in Fig. 1, we train the GPT-2 (Radford et al., 2019) on the WikiText-103 dataset (Merity et al., 2017) by Adam and SGD, respectively. The results demonstrate Adam's superiority over SGD in handling class imbalance. Based on the theoretical insight from Francazi et al. (2023), per-class gradient normalization (Francazi et al., 2023) can mitigate class imbalance by decoupling optimization dynamics from class-level scaling, as shown in Eq. 6. If gradients across classes are orthogonal, the element-wise gradient normalization (as employed in Adam) can functionally approximate per-class normalization by normalizing per-weight gradients, thus mitigating class imbalance, as shown in Eq. 7. For clarity, the weight update rules of the two methods are formulated as follows.

For per-class normalization (Francazi et al., 2023):

$$\theta_{n,t+1} = \theta_{n,t} - \eta \sum_{c=1}^{C} \frac{\alpha^{(c)} \nabla\theta_{n,t}^{(c)}}{\left\|\alpha^{(c)} \nabla\theta_{n,t}^{(c)}\right\|} = \theta_{n,t} - \eta \sum_{c=1}^{C} \frac{\nabla\theta_{n,t}^{(c)}}{\left\|\nabla\theta_{n,t}^{(c)}\right\|}, \quad \sum_{c=1}^{C} \nabla\theta_{n,t}^{(c)} = \nabla\theta_{n,t}, \quad (6)$$

where the $\theta_{n,t}$ denotes the $n^{th}$ model weight at the $t^{th}$ iteration, the $\eta$ represents the learning rate and $C$ indicates the number of classes. The $\alpha^{(c)}$ is the scaling factor of class $c$ while the term $\nabla\theta_{n,t}^{(c)}$ is the $n^{th}$ gradient from class $c$ at the $t^{th}$ iteration. We use $\|\cdot\|$ to denote the $L^2$-normalization over the $n$ gradients. Although Eq. 6 demonstrates scale invariance in inter-class gradient magnitudes, calculating gradients separately for each class significantly increases both time and memory costs.

For Adam's element-wise gradient normalization (Kingma & Ba, 2014), while gradients across classes are orthogonal:

$$\theta_{n,t+1} = \theta_{n,t} - \eta \frac{\alpha^{(c)} \nabla\theta_{n,t}^{(c)}}{\sqrt{EMA((\alpha^{(c)} \nabla\theta_{n,t}^{(c)})^2)/(1-\beta_1^t)}} = \theta_{n,t} - \eta \frac{\nabla\theta_{n,t}^{(c)}}{\sqrt{EMA((\nabla\theta_{n,t}^{(c)})^2)/(1-\beta_1^t)}}, \quad (7)$$

where

$$EMA(\nabla\theta_{n,t}^{(c)2}) = v_{n,t}, \quad v_{n,t} = \beta_1 \cdot v_{n,t-1} + (1-\beta_1) \cdot (\nabla\theta_{n,t}^{(c)})^2, \quad v_{n,0} = 0. \quad (8)$$

Here, the function $EMA(\cdot)$ represents the Exponential Moving Average (EMA), which is used to estimate the second-order moment in Adam, and the scalar $\beta_1$ is a hyperparameter. The other symbols align with Eq. 6.

Eq. 7 demonstrates scale invariance in inter-class gradient magnitudes under orthogonal inter-class gradient conditions. This point aligns with the findings in Kunstner et al. (2024). However, the assumption of fully orthogonal gradients between classes may be too strong. Thus, we challenge the argument that element-wise gradient normalization can mitigate the class imbalance by balancing the element-level gradients.

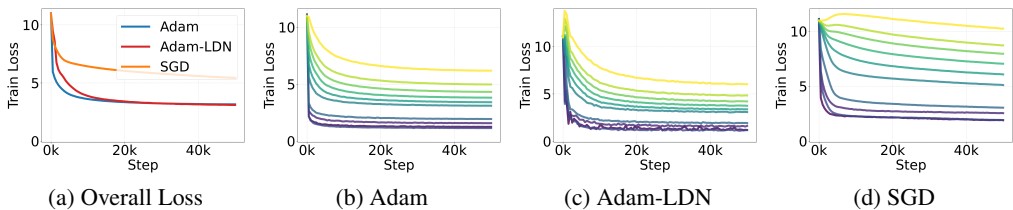

Figure 2: Comparison of Adam, Adam-LDN, and SGD when training GPT-2 on class-imbalanced data (i.e., WikiText-103). Adam-LDN is an ablation variant of Adam that removes the ability to balance gradients across elements. The results suggest that Adam without element-wise normalization (i.e., Adam-LDN) can also achieve comparable or even superior results to Adam on class-imbalanced data.

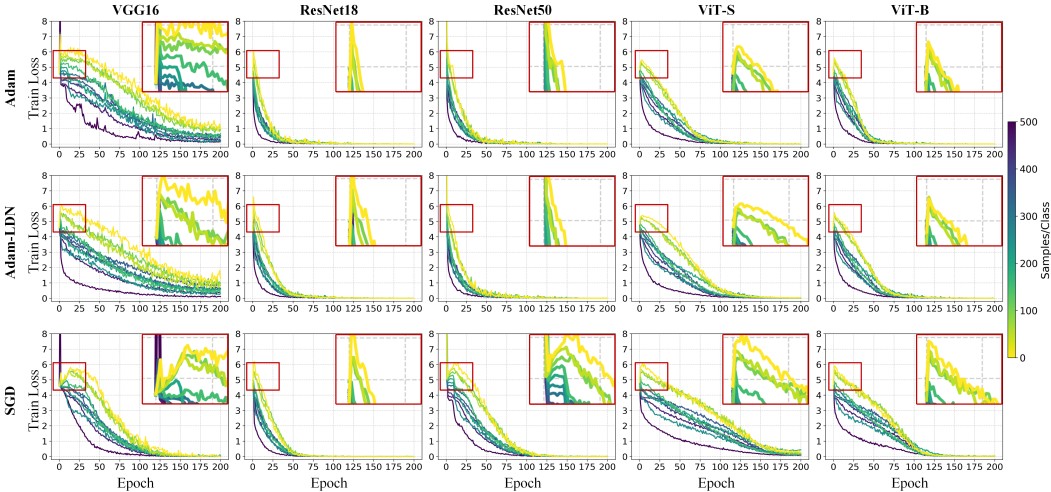

Figure 3: The training loss for 11 selected classes in a class-imbalanced dataset during model training. These classes (Nos. 1, 10, 20, 30, 40, 50, 60, 70, 80, 90, 100) were sampled from a descending numerical ranking of all available classes. The used dataset is a subset of CIFAR100 Krizhevsky et al. (2009) (dubbed CIFAR100-LT for simplicity), where the $c^{th}$ class contains $500 \times \frac{1}{10^{\rho}}$ samples. Here, the $\rho$ is uniformly sampled from the interval [0,1] with its value defined as $\rho = \frac{c}{C}$, where $C$ denotes the total number of classes. This data process follows the previous works (Cao et al., 2019; Cui et al., 2022). The results imply that gradients between classes may not be orthogonal, at least during the initial stages, as Adam fails to mitigate class imbalance.

To investigate this issue, we try to separate the role of balancing the element-wise gradients from Adam and compare the difference before and after the separation, as shown in Figs. 2 and 3. To isolate the effect of balancing the element-wise gradients, we replace the element-wise gradient normalization in Adam (i.e., Eq. 3) with the layer-wise dynamics normalization, and the Adam

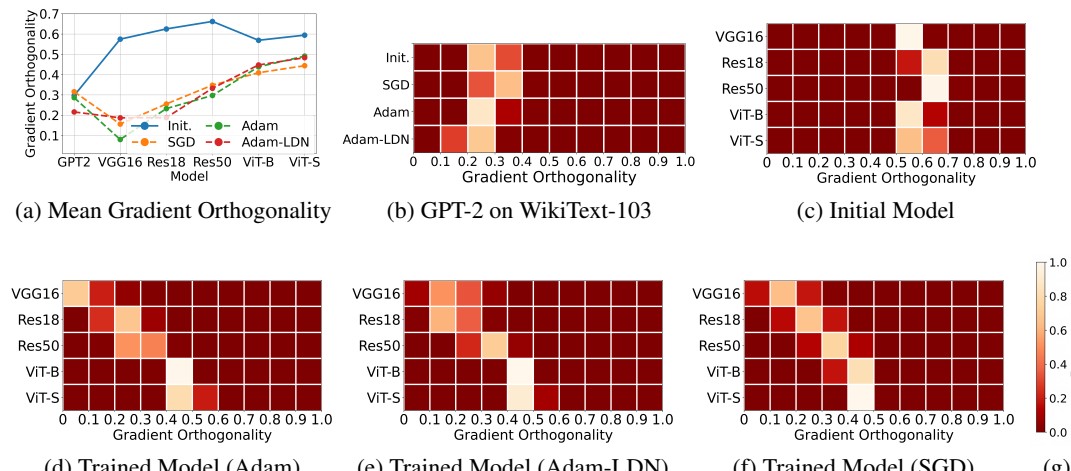

Figure 4: The inter-class gradient orthogonality of models trained by optimizers on WikiText-103 or CIFAR100-LT. **(a)**: The Mean Gradient Orthogonality (MGO) for different visual models trained on WikiText-103 or CIFAR1100-LT. MGO can be calculated by Eq. 10. **(b)**: The Mean Gradient Orthogonality (MGO) for the GPT-2 trained on WikiText-103 with different optimizers. **(c)**: The distribution of gradient orthogonality $GO(x_i, x_j; f_\theta)$ for different data pairs $(x_i, x_j)$ at model initialization. We sample 100 data from different classes and construct 4,950 pairs. **(d)-(f)**: The distribution of gradient orthogonality for trained models. The gradient orthogonality ranges from 0 to 1, where 0 indicates full orthogonality. The results suggest that the inter-class gradients are not orthogonal, particularly during the initial stages of model training.

variant is dubbed Adam-LDN for simplicity. The update of Adam-LDN can be formulated as

$$\theta_{n,t+1}^{(l)} = \theta_{n,t}^{(l)} - lr \cdot \frac{m_{n,t}^{(l)}}{\sqrt{\frac{1}{N^{(l)}} \sum_n \left(m_{n,t}^{(l)}\right)^2}}, \tag{9}$$

where the $m_{n,t}^{(l)}$ is the momentum (in Eq. 4) of the $n^{th}$ weight in layer $l$ at the $t^{th}$ iteration, and the $N^{(l)}$ represents the weight number of layer $l$ in a model.

Compared the results in Fig. 2b with those in Fig. 2c, Element-wise normalization can help Adam better address gradient imbalances caused by class imbalance during the early stages of training. However, after training, both the Adam optimizer and its variants (i.e., Adam-LDN) exhibit nearly identical training losses, effectively coping with class imbalance compared to SGD. Furthermore, during training for the visual task in Fig. 3, element-wise dynamics normalization fails to effectively mitigate gradient imbalances induced by class imbalance in early stages, though these optimizers successfully handle class-imbalanced data throughout the entire training process. The failure suggests that gradients between classes may not be orthogonal, at least during the initial stages of training.

To further demonstrate inter-class gradient orthogonality, Fig. 4 visualizes gradient orthogonality across different classes. The Mean Gradient Orthogonality (MGO) for model $f_\theta$ in Fig. 4a is defined as:

$$MGO(X; f_\theta) = \frac{1}{K} \sum_{x_i, x_j \in X; i \neq j} GO(x_i, x_j; f_\theta), \tag{10}$$

where

$$GO(x_i, x_j; f_\theta) = \frac{1}{N} \sum_l N^{(l)} \cdot Cos(\left|\nabla_{\theta^{(l)}} J(f_\theta(x_i))\right|, \left|\nabla_{\theta^{(l)}} J(f_\theta(x_j))\right|). \tag{11}$$

Here, the $X$ denotes a subset sampled from the original dataset while $x_i$ and $x_j$ are two distinct samples in the subset $X$. The $K$ is the number of the subset $X$, and $N$ is the number of weights in the model $f_\theta$ with its weights $\theta$, while $N^{(l)}$ is the number of the weights in layer $l$. The $Cos(\cdot, \cdot)$

computes cosine similarity. The $\nabla_{\theta^{(l)}} J(f_\theta(x_i))$ represents the weight gradients at layer $l$ w.r.t the loss $J(f_\theta(x_i))$.

As shown in Fig. 4, visual models exhibit low inter-class gradient orthogonality at the beginning of training. This prevents the element-wise dynamics normalization in Adam from effectively handling class-imbalanced data. After training, CNN-based models exhibit high inter-class gradient orthogonality, whereas Transformer-based models show comparatively lower orthogonality.

In summary, Adam effectively handles class-imbalanced data compared to SGD, yet its underlying mechanism diverges from the previous arguments (Kunstner et al., 2024). Crucially, Adam achieves this without element-wise normalization. This raises a pivotal question: What alternative mechanisms enable Adam to mitigate the effects of class imbalance?

## 4.2 The Mechanism of Adam in Class Imbalance Scenarios

Observing Adam in Eq. 3, we identify three roles of dynamics normalization including element-wise dynamics normalization, layer-wise dynamics normalization and inter-iteration dynamics normalization. Specifically, 1) the element-wise dynamics normalization stabilizes individual weight gradients, improving the invariance of optimization dynamics to scaling of any single weight's gradient; 2) the layer-wise dynamics normalization stabilizes gradients within a layer, preventing layer-specific gradient scaling from altering optimization dynamics; and 3) inter-iteration dynamics normalization stabilizes gradients across iterations, trying to decouple optimization dynamics from iteration-dependent gradient scaling.

By modifying Adam in Eq. 3 into Adam-LDN in Eq. 9, Adam-LDN retains two mechanisms of dynamics normalization, which are layer-wise dynamics normalization and inter-iteration dynamics normalization. Compared to Adam, 1) enhanced layer-wise dynamics normalization entirely prevents layer-specific gradient scaling from altering optimization dynamics; and 2) weakened inter-iteration dynamics normalization is limited to decoupling the optimization of dynamics from iteration-dependent layer-wise gradient scaling. This behavior can be formalized in the update rule:

$$\theta_{n,t+1}^{(l)} = \theta_{n,t}^{(l)} - lr \cdot \frac{\alpha_t^{(l)} m_{n,t}^{(l)}}{\sqrt{\frac{1}{N^{(l)}} \sum_n \left(\alpha_t^{(l)} m_{n,t}^{(l)}\right)^2}} = \theta_t^{(l)} - lr \cdot \frac{m_{n,t}^{(l)}}{\sqrt{\frac{1}{N^{(l)}} \sum_n \left(m_{n,t}^{(l)}\right)^2}}, \qquad (12)$$

where the $m_{n,t}^{(l)}$ is the momentum (in Eq. 4) of the $n^{th}$ weight in layer $l$ at the $t^{th}$ iteration. The scaling factor $\alpha_t^{(l)}$ varies per iteration $t$ and layer $l$. The scaling factor $\alpha_t^{(l)}$ cancels out, demonstrating the elimination of iteration-dependent scaling effects.

This property may mitigate class imbalance even under strong gradient coupling. Consider a binary classification task with high-frequency class $c_1$ and low-frequency class $c_2$. Initially, the optimization dynamics are dominated by the gradients of the high-frequency class $c_1$:

$$\|Dynamics_t^{(l)}\|^2 \approx \sum_{n'} \left( \frac{-lr \cdot m_{n',t}^{(c_1,l)}}{\sqrt{\frac{1}{N^{(l)}} \sum_n \left(m_{n,t}^{(c_1,l)}\right)^2}} \right)^2 = lr^2 \cdot N^{(l)}, \quad J(f_\theta(X^{(c_1)})) \gg J(f_\theta(X^{(c_2)})). \tag{13}$$

As the model $f_\theta$ achieves high fitting accuracy on the high-frequency class $c_1$ (i.e., $J(f_\theta(X^{(c_1)})) \to 0$), the relative contribution of the low-frequency class $c_2$ progressively dominates the weight updates:

$$\|Dynamics_t^{(l)}\|^2 \approx \sum_{n'} \left( \frac{-lr \cdot m_{n',t}^{(c_2,l)}}{\sqrt{\frac{1}{N^{(l)}} \sum_n \left(m_{n,t}^{(c_2,l)}\right)^2}} \right)^2 = lr^2 \cdot N^{(l)}, \quad J(f_\theta(X^{(c_1)})) \ll J(f_\theta(X^{(c_2)})). \tag{14}$$

where $\|Dynamics_t^{(l)}\|$ represents the $L^2$ normalization of optimization dynamics, and the $c_1$ and $c_2$ denote high-frequency class and low-frequency one, respectively. The $J(f_\theta(X^{(c_1)}))$ represents

the loss of high-frequency class. Therefore, Adam-LDN's dynamics normalization promotes convergence of the ratio of the losses from different classes toward 1, which can be formulated as

$$J(f_\theta(X^{(c_1)}))/J(f_\theta(X^{(c_2)})) \to 1. \tag{15}$$

This mitigates class imbalance, as supported by the results in Figs. 2 and 3.

In summary, inter-iteration layer-wise dynamics normalization can mitigate class imbalance by balancing the magnitude of optimization dynamics across different classes, while reducing memory overhead equivalent to the model's gradient storage, as the second-order moment $v$ in Adam is omitted.

### 4.3 THE NECESSITY OF LAYER-WISE DYNAMICS NORMALIZATION

Layer-wise dynamics normalization is necessary because normalization layers (e.g., Ioffe & Szegedy (2015); Ba et al. (2016); Wu & He (2018)) introduce inconsistencies at the layer level between forward and backward propagation. These propagation mismatches may result in extremely small gradient-to-weight ratios, severely impeding optimization. Similar inconsistencies may arise from components with inherent normalization properties, such as the attention layer Vaswani et al. (2017) and the softmax function. Specifically, we take the normalization layer as an example.

If we scale the model weights $\theta^{(1)}$ of one layer by the scaling factor $\alpha^{(1)}$, the forward propagation in a typical model with normalization layers can be formulated as

$$\delta(norm(f^{(k)} \cdots f^{(2)} \circ f^{(1)}(x^{(1)}; \theta^{(1)}))) = \delta(norm(f^{(k)} \cdots f^{(2)} \circ f^{(1)}(x^{(1)}; \alpha^{(1)}\theta^{(1)}))), \tag{16}$$

where $f^{(k)}$ denotes the $k^{th}$ operator (e.g., convolutional layer, linear layer), the $x^{(1)}$ is the input of the operator $f^{(1)}$, and $\theta^{(1)}$ represents its weights. The scalar $\alpha^{(1)}$ is a scaling factor. The function $norm(\cdot)$ denotes a normalization layer, and $\delta(\cdot)$ is the activation function or any other function. Crucially, scaling $\theta^{(1)}$ by $\alpha^{(1)}$ preserves the output in Eq. 16 due to the intrinsic rescaling property of normalization operations. This demonstrates functional invariance under weight scaling.

However, while scaling the weights as shown in Eq. 16, the corresponding gradient in back-propagation is as follows:

$$\frac{\partial\delta(norm(f^{(k)} \cdots f^{(2)} \circ f^{(1)}(x^{(1)}; \theta^{(1)})))}{\partial\left(\alpha^{(1)}\theta^{(1)}\right)} = \frac{1}{\alpha^{(1)}} \frac{\partial\delta(norm(f^{(k)} \cdots f^{(2)} \circ f^{(1)}(x^{(1)}; \alpha^{(1)}\theta^{(1)})))}{\partial\theta^{(1)}}. \tag{17}$$

As Eq. 17 demonstrates, although model functionality remains invariant under weight scaling, the gradients for this layer are scaled by $1/\alpha^{(1)}$. This establishes an inverse proportionality: a larger scaling factor reduces the magnitudes of the gradient within the layer. Such inverse proportionality risks vanishing gradients and imbalanced update step sizes across layers, invalidating inter-layer gradient comparisons due to heterogeneous scaling effects. This is also why gradient orthogonality in Eq. 10 requires layer-by-layer computations. However, whether employing element-level dynamics normalization or layer-level dynamics normalization, both only balance dynamics differences without compensating for weight discrepancies. This can be formulated as

$$\begin{aligned} \theta^{(1)}_{n,t+1} &= \alpha^{(1)} \cdot \theta^{(1)}_{n,t} - lr \cdot Norm(\nabla\theta^{(1)}_{n,t}/\alpha^{(1)}) = \alpha^{(1)} \cdot \theta^{(1)}_{n,t} - lr \cdot Norm(\nabla\theta^{(1)}_{n,t}) \\ &\neq \theta^{(1)}_{n,t} - lr \cdot Norm(\nabla\theta^{(1)}_{n,t}) \end{aligned}. \tag{18}$$

Eq. 18 implies that the imbalanced initialization may lead to unfair optimization. To harmonize optimization dynamics, we introduce layer-specific scaling factors that are proportional to initial weight magnitudes in the layer. The Adam optimizer with layer-specific scaling factors is referred to as Adam-S, while the Adam-LDN with layer-specific scaling factors is referred to as Adam-S-LDN. As demonstrated in Figs. 5 and 6, this approach effectively improves the convergence of models, especially for the VGG16.

In summary, while models with normalization layers are functionally invariant to layer-level weight scaling, this scaling can nonetheless slow down model optimization. Layer-wise dynamics normalization can mitigate this issue by decoupling gradient magnitudes from scaling effects. However, the update rate depends on the weight-to-dynamics ratio. Although optimization dynamics stabilize, weight imbalances may still impede optimization. We address this limitation through adaptive scaling factors that prevent initialization-induced discrepancies.

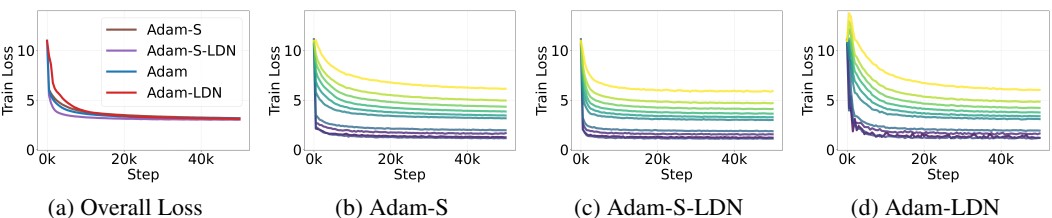

(a) Overall Loss  (b) Adam-S  (c) Adam-S-LDN  (d) Adam-LDN

Figure 5: Comparison of Adam-S, Adam-S-LDN and Adam-LDN when training GPT-2 on class-imbalanced data (i.e., WikiText-103).

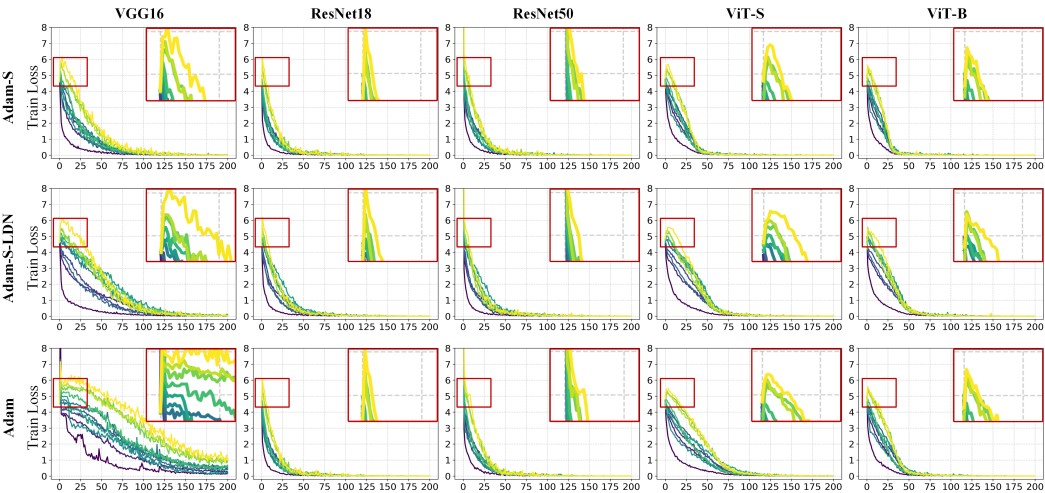

Figure 6: The training loss for 11 selected classes in a class-imbalanced dataset during model training. These classes (Nos. 1, 10, 20, 30, 40, 50, 60, 70, 80, 90, 100) were sampled from a descending numerical ranking of all available classes. The models are trained on CIFAR100-LT used in Fig. 4.

## 5 DISCUSSION AND LIMITATIONS

Our work primarily investigates the interpretability of Adam's mechanisms, rather than developing novel optimizers, which may potentially enable the evolution of an optimizer's advantages over Adam and other optimizers. In addition, our work mainly focuses on model training performance rather than generalization, as poor training performance inevitably leads to poor generalization. From another perspective, this work proposes a novel exploration pathway to improve Adam-S-LDN without increasing storage costs. For instance, constraining the distribution form of the optimization momentum within a single layer. In the era of large models, the reduction in storage expenditure also signifies potential performance gains.

## 6 CONCLUSIONS

This work reveals that Adam mitigates class imbalance by stabilizing optimization dynamics across different iterations. We also demonstrate that layer-level gradients across iterations are the coarsest feasible granularity for this operation. Element-wise dynamics normalization primarily accelerates convergence, with its efficacy contingent on gradient orthogonality, which is influenced by multiple factors, including task specifications and model architecture. Consequently, a scaling factor proportional to the initialized weight magnitudes can be introduced to further harmonize layer-wise optimization step sizes. Also, we can try to reduce the memory expenditure of the full gradient size by replacing element-wise dynamics normalization with layer-wise dynamics normalization.

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

# A    APPENDIX

## CONTENTS

## A.1    THE USE OF LARGE LANGUAGE MODELS

We employ large language models to check for typos and grammatical errors, and utilize the LLMs to provide suggestions for fixing code bugs.

## A.2    EXPERIMENT DETAILS

### A.2.1    DATASETS

- **WikiText-103** (Merity et al., 2017) using sequences of 1,024 tokens and the BPE tokenizer (Sennrich et al., 2016), with a vocabulary of size 50,257.
- **CIFAR-100-LT** This is a subset of CIFAR100 Krizhevsky et al. (2009), where the $c^{th}$ class contains $500 \times \frac{1}{10^\rho}$ samples by default. This refers to a situation where the imbalance ratio is 1/10. If the imbalance ratio is 1/20, the $c^{th}$ class contains $500 \times \frac{1}{20^\rho}$ samples. Here,

the $\rho$ is uniformly sampled from the interval [0,1] with its value defined as $\rho = \frac{c}{C}$, where $C$ denotes the total number of classes. This data process follows the previous works (Cao et al., 2019; Cui et al., 2022).

- **The Heavy Tailed ImageNet** This is a subset of ImageNet (Russakovsky et al., 2015), subsampled to exhibit class imbalance. We sample $\left\lceil \frac{1300}{k} \right\rceil$ images from the $k^{th}$ class. The process for generating this ImageNet subset for evaluation follows that of the work (Kunstner et al., 2024).

### A.2.2 MODELS

- **GPT-2** (Radford et al., 2019) We follow the setup in (Radford et al., 2019). The embedding dimension is 768.
- **VGG16** (Simonyan & Zisserman, 2014) is a CNN-based model. We use the variant with batch normalization layers.
- **ResNet** (He et al., 2016) is a CNN-based model. ResNets follow the setting of ResNet in (Chen et al., 2020) when training on CIFAR-100. We replace the first 7x7 convolution of stride 2 with a 3x3 convolution of stride 1 and remove the first max pooling operation. We train the vanilla ResNets on the ImageNet dataset.
- **ViT-S** (Dosovitskiy et al., 2020) is a Transformer-based model. For images sampled from the CIFAR100-LT dataset, we split each image into 4×4 patches as tokens. For the ImageNet dataset, we split each image into 16×16 patches as tokens. The embedding dimension is 384, and the number of heads is 6.
- **ViT-B** (Dosovitskiy et al., 2020) is a Transformer-based model. For images sampled from the CIFAR100-LT dataset, we split each image into 4×4 patches as tokens. For the ImageNet dataset, we split each image into 16×16 patches as tokens. The embedding dimension is 768, and the number of heads is 12.

### A.2.3 OPTIMIZATION ALGORITHMS

In the algorithms, for simplicity, although $\theta^{(l)}$ is a tensor, we denote one of its elements as $\theta^{(l)}_{n,t}$. By default, all formulas are depicted as operations at the elemental level, i.e., calculations based on scalars. We employ the cosine scheduler (Loshchilov & Hutter, 2017) to adjust the learning rate during model training.

- **SGD** Stochastic Gradient Descent simply scales the gradients by the step size as the updated values, providing noisy optimization dynamics that come from mini-batches. The details are shown in Alg. 1.
- **Adam** (Kingma & Ba, 2014) is a prevalent optimizer, which combines the ideas of momentum and RMSProp (Tieleman et al., 2012). The details are shown in Alg. 2.
- **Adam-LDN**, an Adam variant, replaces element-wise dynamics normalization with layer-wise dynamics normalization. Unlike the element-wise normalization, which requires storing first-order moments for each model weight, the layer-wise method eliminates this storage requirement. The details are shown in Alg. 3.
- **Adam-S** is the Adam variant, which introduces a scaling factor to harmonize layer-level optimization step sizes further. The details are shown in Alg. 4.
- **Adam-S-LDN**, an Adam-S variant, replaces element-wise dynamics normalization with layer-wise dynamics normalization. Unlike the element-wise normalization, which requires storing second-order moments for each model weight, the layer-wise method eliminates this storage requirement. The details are shown in Alg. 5.

### A.2.4 SUMMARY OF SETTINGS USED

For clarity, this section presents the settings used in the experiments described above. The details are listed in Table 1. Also, the test performance is shown in Table 1. We run all the experiments on 4 V100 GPUs.

For all the model training, we employ the cosine scheduler (Loshchilov & Hutter, 2017) to adjust the learning rate during model training.

---

**Algorithm 1** Stochastic Gradient Descent.

---

1: **Input:** Initial weights $\theta_{n,0}$, learning rate schedule $\{lr_t\}_{t=0}^T$, datasets $D_{train}$.
2: **for** $t = 1$ **to** $T$ **do**
3:    Sample mini-batch of data $d \sim D_{train}$ and calculate the gradients $\nabla\theta_{n,t}$.
4:    Update the weights $\theta_{n,t}^{(l)} \leftarrow \theta_{n,t-1}^{(l)} - lr_t \cdot \nabla\theta_{n,t}^{(l)}$.
5: **end for**

---

**Algorithm 2** Adam.

---

1: **Input:** Initial weights $\theta_{n,0}$, learning rate schedule $\{lr_t\}_{t=0}^T$, datasets $D_{train}$, hyper-parameter $\beta_1, \beta_2$, variable $m, v$.
2: Initialize $m_{n,t}^{(l)} \leftarrow 0$, $v_{n,t}^{(l)} \leftarrow 0$.
3: **for** $t = 1$ **to** $T$ **do**
4:    Sample mini-batch of data $d \sim D_{train}$ and calculate the gradients $\nabla\theta_{n,t}$.
5:    Update the variable $m_{n,t}^{(l)} = \beta_2 \cdot m_{n,t-1}^{(l)} + (1 - \beta_2) \cdot \nabla\theta_{n,t}^{(l)}$.
6:    Update the variable $v_{n,t}^{(l)} = \beta_1 \cdot v_{n,t-1}^{(l)} + (1 - \beta_1) \cdot (\nabla\theta_{n,t}^{(l)})^2$.
7:    Update the weights $\theta_{n,t}^{(l)} \leftarrow \theta_{n,t-1}^{(l)} - lr_t \cdot \frac{m_{n,t}^{(l)}/(1-\beta_2^t)}{\sqrt{v_{n,t}^{(l)}/(1-\beta_1^t)}}$.
8: **end for**

---

**Algorithm 3** Adam-LDN.

---

1: **Input:** Initial weights $\theta_{n,0}$, learning rate schedule $\{lr_t\}_{t=0}^T$, datasets $D_{train}$, hyper-parameter $\beta_2$, variable $m$.
2: Initialize $m_{n,t}^{(l)} \leftarrow 0$.
3: **for** $t = 1$ **to** $T$ **do**
4:    Sample mini-batch of data $d \sim D_{train}$ and calculate the gradients $\nabla\theta_{n,t}$.
5:    Update the variable $m_{n,t}^{(l)} = \beta_2 \cdot m_{n,t-1}^{(l)} + (1 - \beta_2) \cdot \nabla\theta_{n,t}^{(l)}$.
6:    Update the weights $\theta_{n,t}^{(l)} \leftarrow \theta_{n,t-1}^{(l)} - lr_t \cdot \frac{m_{n,t}^{(l)}}{\sqrt{\frac{1}{N}\sum_n (m_{n,t}^{(l)})^2}}$.
7: **end for**

---

**Algorithm 4** Adam-S.

---

1: **Input:** Initial weights $\theta_{n,0}$, learning rate schedule $\{lr_t\}_{t=0}^T$, datasets $D_{train}$, hyper-parameter $\beta_1, \beta_2$, variable $m, v$.
2: Initialize $m_{n,t}^{(l)} \leftarrow 0$, $v_{n,t}^{(l)} \leftarrow 0$, $\alpha^{(l)} \leftarrow \sqrt{\frac{1}{N}\sum_n (\theta_{n,0}^{(l)})^2}$.
3: Initialize the $\alpha^{(l)}$ for the bias $\alpha^{(l)} \leftarrow \alpha \cdot \alpha^{(l-1)}$.
4: **for** $t = 1$ **to** $T$ **do**
5:    Sample mini-batch of data $d \sim D_{train}$ and calculate the gradients $\nabla\theta_{n,t}$.
6:    Update the variable $m_{n,t}^{(l)} = \beta_2 \cdot m_{n,t-1}^{(l)} + (1 - \beta_2) \cdot \nabla\theta_{n,t}^{(l)}$.
7:    Update the variable $v_{n,t}^{(l)} = \beta_1 \cdot v_{n,t-1}^{(l)} + (1 - \beta_1) \cdot (\nabla\theta_{n,t}^{(l)})^2$.
8:    Update the weights $\theta_{n,t}^{(l)} \leftarrow \theta_{n,t-1}^{(l)} - lr_t \cdot \alpha^{(l)} \cdot \frac{m_{n,t}^{(l)}/(1-\beta_2^t)}{\sqrt{v_{n,t}^{(l)}/(1-\beta_1^t)}}$.
9: **end for**

---

## A.3 RELATIONSHIP BETWEEN LEARNING RATE AND PERFORMANCE

To demonstrate the relationship between learning rate and performance across different optimizers, we search for the optimal learning rate and plot the performance of the optimal point and its six neighbouring points, as shown in Fig. 7. Note that this observation does not indicate the optimization algorithm's robustness to changes in the learning rate, as a unit change in the learning rate will result

---

**Algorithm 5** Adam-S-LDN.

---

1: **Input:** Initial weights $\theta_{n,0}$, learning rate schedule $\{lr_t\}_{t=0}^T$, datasets $D_{train}$, hyper-parameter $\beta_2$, variable $m$.

2: Initialize $m_{n,t}^{(l)} \leftarrow 0$, $v_{n,t}^{(l)} \leftarrow 0$, $\alpha^{(l)} \leftarrow \sqrt{\frac{1}{N} \sum_n (\theta_{n,0}^{(l)})^2}$.

3: Initialize the $\alpha^{(l)}$ for the bias $\alpha^{(l)} \leftarrow \alpha \cdot \alpha^{(l-1)}$.

4: **for** $t = 1$ **to** $T$ **do**

5:     Sample mini-batch of data $d \sim D_{train}$ and calculate the gradients $\nabla\theta_{n,t}$.

6:     Update the variable $m_{n,t}^{(l)} = \beta_2 \cdot m_{n,t-1}^{(l)} + (1 - \beta_2) \cdot \nabla\theta_{n,t}^{(l)}$.

7:     Update the weights $\theta_{n,t}^{(l)} \leftarrow \theta_{n,t-1}^{(l)} - lr_t \cdot \alpha^{(l)} \cdot \frac{m_{n,t}^{(l)}}{\sqrt{\frac{1}{N} \sum_n (m_{n,t}^{(l)})^2}}$.

8: **end for**

---

Table 1: Summary of models, dataset, optimizers, and hyperparameters used.

| Model | Dataset | Optimizer | Batch size | Learning rate | Metrics | Test performance |
|---|---|---|---|---|---|---|
| GPT-2 | WikiText-103 | SGD | 32 | $5 \times 10^{-4}$ | Perplexity | 64.88 |
| GPT-2 | WikiText-103 | Adam | 32 | $5 \times 10^{-5}$ | Perplexity | 22.20 |
| GPT-2 | WikiText-103 | Adam-LDN | 32 | $5 \times 10^{-5}$ | Perplexity | 21.01 |
| GPT-2 | WikiText-103 | Adam-S | 32 | $5 \times 10^{-3}$ | Perplexity | 18.78 |
| GPT-2 | WikiText-103 | Adam-S-LDN | 32 | $5 \times 10^{-2}$ | Perplexity | 19.26 |
| VGG16 | CIFAR-100-LT | All | 256 | see Fig. 7 | Accuracy | see Fig. 7 |
| ResNet18 | CIFAR-100-LT | All | 256 | see Fig. 7 | Accuracy | see Fig. 7 |
| ResNet50 | CIFAR-100-LT | All | 256 | see Fig. 7 | Accuracy | see Fig. 7 |
| ViT-S | CIFAR-100-LT | All | 256 | see Fig. 7 | Accuracy | see Fig. 7 |
| ViT-B | CIFAR-100-LT | All | 256 | see Fig. 7 | Accuracy | see Fig. 7 |

in different magnitudes of change in the update step size across different optimization algorithms. For example, comparing Adam (Algorithm 2) with Adam-S (Algorithm 4):

- Adam scales the term $\frac{m_{n,t}^{(l)}/(1-\beta_2^t)}{\sqrt{v_{n,t}^{(l)}/(1-\beta_1^t)}}$ by the learning rate $lr_t$;

- Adam-S scales the same term by the factor $lr_t \cdot \alpha^{(l)}$.

In fact, the scaling factor $lr_t \cdot \alpha^{(l)}$ in Adam-S is functionally equivalent to the learning rate $lr_t$ in Adam. Still, the scaling factors $lr_t \cdot \alpha^{(l)}$ of different layers are different while the learning rate $lr_t$ is constant across various layers.

A.4 DERIVATION OF EQUATION 7

If gradients across classes are orthogonal, the element-wise gradient normalization (as employed in Adam) can functionally approximate per-class normalization by normalizing per-weight gradients, thus mitigating class imbalance, as shown in Eq. 19.

For Adam's element-wise gradient normalization (Kingma & Ba, 2014), while gradients across classes are orthogonal:

$$\theta_{n,t+1} = \theta_{n,t} - \eta \frac{\nabla\theta_{n,t}^{(c)}}{\sqrt{EMA((\nabla\theta_{n,t}^{(c)})^2)/(1 - \beta_1^t)}}, \tag{19}$$

where

$$EMA(\nabla\theta_{n,t}^{(c)^2}) = v_{n,t}, \quad v_{n,t} = \beta_1 \cdot v_{n,t-1} + (1 - \beta_1) \cdot (\nabla\theta_{n,t}^{(c)})^2, \quad v_{n,0} = 0. \tag{20}$$

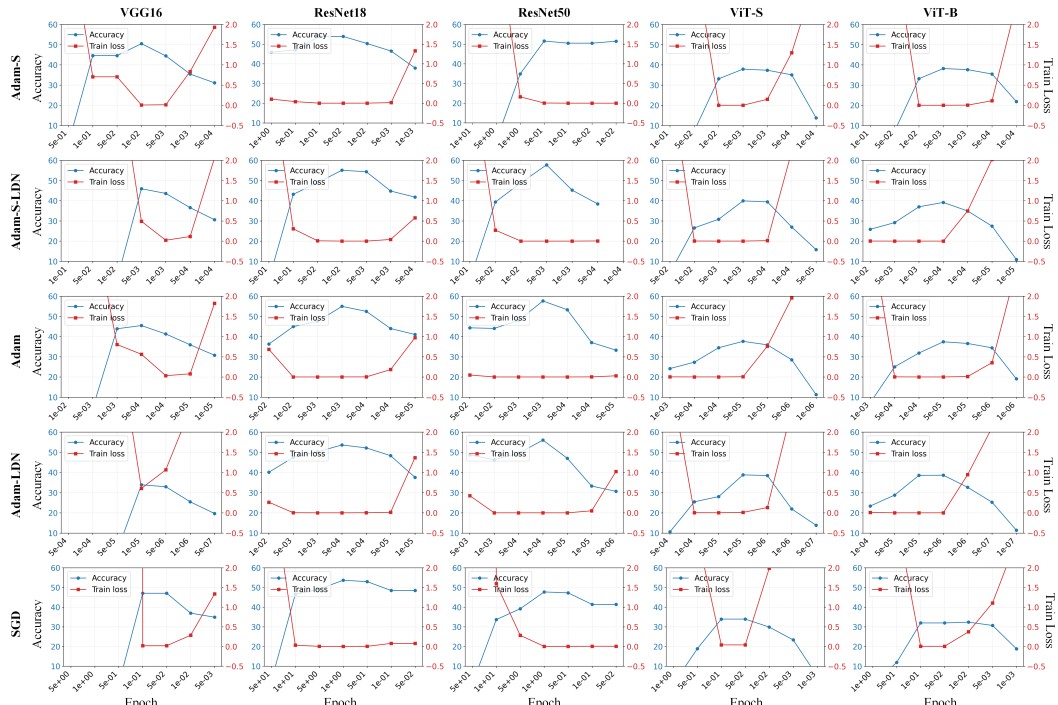

Figure 7: Relationship between Training losses and accuracies for models trained on CIFAR100-LT with different optimizers. Note that this observation does not indicate the optimization algorithm's robustness to changes in the learning rate, as a unit change in the learning rate will result in different magnitudes of change in the update step size across different optimization algorithms. For example, comparing Adam (Algorithm 2) with Adam-S (Algorithm 4), Adam scales the term $\frac{m_{n,t}^{(l)}/(1-\beta_2^t)}{\sqrt{v_{n,t}^{(l)}/(1-\beta_1^t)}}$ by the learning rate $lr_t$ while Adam-S scales the same term by the factor $lr_t \cdot \alpha^{(l)}$. In fact, the scaling factor $lr_t \cdot \alpha^{(l)}$ in Adam-S is functionally equivalent to the learning rate $lr_t$ in Adam.

Here, the $\theta_{n,t}$ denotes the $n^{th}$ model weight at the $t^{th}$ iteration, the $\eta$ represents the learning rate and $C$ indicates the number of classes. The $\alpha^{(c)}$ is the scaling factor of class $c$ while the term $\nabla\theta_{n,t}^{(c)}$ is the $n^{th}$ gradient from class $c$ at the $t^{th}$ iteration. The function $EMA(\cdot)$ represents the Exponential Moving Average (EMA), which is used to estimate the second-order moment in Adam, and the scalar $\beta_1$ is a hyperparameter.

If the gradient $\nabla\theta_{n,t}^{(c)}$ from class $c$ is scaled by $\alpha^{(c)}$, Eq. 19 can be written as

$$\theta_{n,t+1} = \theta_{n,t} - \eta \frac{\alpha^{(c)}\nabla\theta_{n,t}^{(c)}}{\sqrt{EMA((\alpha^{(c)}\nabla\theta_{n,t}^{(c)})^2)/(1-\beta_1^t)}}. \tag{21}$$

Since gradients across different classes are orthogonal, we have the following:

$$\begin{aligned} EMA((\alpha^{(c)}\nabla\theta_{n,t}^{(c)})^2) &= \beta_1 \cdot (\alpha^{(c)})^2 v_{n,t-1} + (1-\beta_1) \cdot (\alpha^{(c)}\nabla\theta_{n,t}^{(c)})^2 \\ &= (\alpha^{(c)})^2 v_{n,t} = (\alpha^{(c)})^2 EMA((\nabla\theta_{n,t}^{(c)})^2), \quad (\alpha^{(c)})^2 v_{n,0} = 0 \end{aligned} \tag{22}$$

Therefore, Eq. 21 can be written as

$$
\begin{aligned}
\theta_{n,t+1} &= \theta_{n,t} - \eta \frac{\alpha^{(c)} \nabla \theta_{n,t}^{(c)}}{\sqrt{EMA((\alpha^{(c)} \nabla \theta_{n,t}^{(c)})^2)/(1-\beta_1^t)}} \\
&= \theta_{n,t} - \eta \frac{\alpha^{(c)} \nabla \theta_{n,t}^{(c)}}{\alpha^{(c)} \sqrt{EMA((\nabla \theta_{n,t}^{(c)})^2)/(1-\beta_1^t)}}, \\
&= \theta_{n,t} - \eta \frac{\nabla \theta_{n,t}^{(c)}}{\sqrt{EMA((\nabla \theta_{n,t}^{(c)})^2)/(1-\beta_1^t)}}
\end{aligned}
\tag{23}
$$

This demonstrates scale invariance in inter-class gradient magnitudes under orthogonal inter-class gradient conditions.

## A.5 CONVERGENCE ANALYSIS FOR ADAM-S-LDN

We now provide convergence analysis for Adam-S-LDN. We assume the function $J(f(x;\theta))$ is $L_l$-smooth with respect to the weights of $l$-th layer $\theta^{(l)}$. There exists a constant $L_l$ such that

$$
\|\nabla_{\theta^{(l)}} J(f(x;\theta)) - \nabla_{\theta'^{(l)}} J(f(x;\theta'))\| \le L_l \left\| \theta^{(l)} - \theta'^{(l)} \right\|, \quad x \in X.
\tag{24}
$$

Let the learning rate $\eta_t = \eta$ for all $\{\eta\}_{t=0}^{T-1}$ and the $\beta_2$ is 0, then we have the following bound:

$$
\begin{aligned}
&\frac{1}{T} \sum_{t,l} \mathbb{E}\left[ \|\nabla J \circ f_{\theta_t}\| \right] \\
&\le \frac{J \circ f_{\theta_0} - J \circ f_{\theta_{opt}}}{T\eta\alpha_{min}} + \frac{2\alpha_{max} \sum_{t,l} \mathbb{E}\left[ \left\| \sum_b \left( \nabla \theta_t^{(l)} - \mathbb{E}\left[\nabla \theta_t^{(l)}\right] \right) \right\| \right]}{\alpha_{min} Tb} + \frac{\eta\alpha_{max}^2 \|L\|_1}{2\alpha_{min}},
\end{aligned}
\tag{25}
$$

where $J$ and $f_{\theta_t}$ represent the loss function and model with weights $\theta_t$. The $\nabla\theta$ denotes the gradients of a single sample. The $\theta_{opt}$ represents the optimal weights. The $\alpha_{min}$ and $\alpha_{min}$ are the minimum and maximum values of the scaling factors $\sqrt{N^{(l)}}\alpha^{(l)}$. The $b$ is batch size. By adjusting the iteration number $T$, batch size $b$, and learning rate $\eta$, the upper bound on the convergence rate can be reduced.

*Proof.* We analyze the convergence of Adam-S-LDN under general mini-batch sizes. For simplicity, we analyze the following update of Adam-S-LDN where $\beta_2$ is 0:

$$
\theta_{n,t}^{(l)} = \theta_{n,t-1}^{(l)} - \eta \cdot \alpha^{(l)} \cdot \frac{m_{n,t}^{(l)}}{\sqrt{\frac{1}{N^{(l)}} \sum_n \left(m_{n,t}^{(l)}\right)^2}}.
\tag{26}
$$

Assuming that the function $J(f_\theta(x))$ is $L_l$-smooth with respect to the weights of $l$-th layer $\theta^{(l)}$ we have the following:

$$
\begin{aligned}
J \circ f_{\theta_{t+1}} &\le J \circ f_{\theta_t} + \langle \nabla_{\theta_t^{(l)}} J \circ f_{\theta_t}, \theta_{t+1}^{(l)} - \theta_t^{(l)} \rangle + \sum_l \frac{L_l}{2} \left\| \theta_{t+1}^{(l)} - \theta_t^{(l)} \right\|^2 \\
&= J \circ f_{\theta_t} - \eta \sum_l \sqrt{N} \alpha^{(l)} \langle \nabla_{\theta_t^{(l)}} J \circ f_{\theta_t}, \frac{m_t^{(l)}}{\left\| m_t^{(l)} \right\|} \rangle + \frac{\eta^2 \left\| NL\alpha^2 \right\|_1}{2}
\end{aligned}
\tag{27}
$$

$$J \circ f_{\theta_{t+1}} \leq J \circ f_{\theta_t} - \eta \sum_l \sqrt{N^{(l)}} \alpha^{(l)} \left\| \nabla_{\theta_t^{(l)}} J \circ f_{\theta_t} \right\|$$

$$- \eta \sum_l \sqrt{N^{(l)}} \alpha^{(l)} \left( \langle \nabla_{\theta_t^{(l)}} J \circ f_{\theta_t}, \frac{m_t^{(l)}}{\left\| m_t^{(l)} \right\|} \rangle - \left\| \nabla_{\theta_t^{(l)}} J \circ f_{\theta_t} \right\| \right) + \frac{\eta^2 \left\| NL\alpha^2 \right\|_1}{2}$$

$$= J \circ f_{\theta_t} - \eta \sum_l \sqrt{N^{(l)}} \alpha^{(l)} \left\| \nabla_{\theta_t^{(l)}} J \circ f_{\theta_t} \right\| + \frac{\eta^2 \left\| NL\alpha^2 \right\|_1}{2}$$

$$- \frac{\eta}{\left\| m_t^{(l)} \right\|} \sum_l \sqrt{N^{(l)}} \alpha^{(l)} \left( \left\| m_t^{(l)} \right\|^2 - \langle m_t^{(l)} - \nabla_{\theta_t^{(l)}} J \circ f_{\theta_t}, m_t^{(l)} \rangle - \left\| m_t^{(l)} \right\| \left\| \nabla_{\theta_t^{(l)}} J \circ f_{\theta_t} \right\| \right)$$

$$\leq J \circ f_{\theta_t} - \eta \sum_l \sqrt{N^{(l)}} \alpha^{(l)} \left\| \nabla_{\theta_t^{(l)}} J \circ f_{\theta_t} \right\| + \frac{\eta^2 \left\| NL\alpha^2 \right\|_1}{2}$$

$$- \eta \sum_l \sqrt{N^{(l)}} \alpha^{(l)} \left( \left\| m_t^{(l)} \right\| - \left\| m_t^{(l)} - \nabla_{\theta_t^{(l)}} J \circ f_{\theta_t} \right\| - \left\| \nabla_{\theta_t^{(l)}} J \circ f_{\theta_t} \right\| \right)$$

$$\leq J \circ f_{\theta_t} - \eta \sum_l \sqrt{N^{(l)}} \alpha^{(l)} \left\| \nabla_{\theta_t^{(l)}} J \circ f_{\theta_t} \right\| + \frac{\eta^2 \left\| NL\alpha^2 \right\|_1}{2}$$

$$- 2\eta \sum_l \sqrt{N^{(l)}} \alpha^{(l)} \left( \left\| m_t^{(l)} \right\| - \left\| m_t^{(l)} - \nabla_{\theta_t^{(l)}} J \circ f_{\theta_t} \right\| - \left\| \nabla_{\theta_t^{(l)}} J \circ f_{\theta_t} \right\| \right)$$

$$\leq J \circ f_{\theta_t} - \eta \sum_l \sqrt{N^{(l)}} \alpha^{(l)} \left\| \nabla_{\theta_t^{(l)}} J \circ f_{\theta_t} \right\| + \frac{\eta^2 \left\| NL\alpha^2 \right\|_1}{2}$$

$$- 2\eta \sum_l \sqrt{N^{(l)}} \alpha^{(l)} \left( \left\| m_t^{(l)} \right\| - \left\| m_t^{(l)} - \nabla_{\theta_t^{(l)}} J \circ f_{\theta_t} \right\| - \left\| m_t^{(l)} \right\| - \left\| \nabla_{\theta_t^{(l)}} J \circ f_{\theta_t} - m_t^{(l)} \right\| \right)$$

$$= J \circ f_{\theta_t} - \eta \sum_l \sqrt{N^{(l)}} \alpha^{(l)} \left\| \nabla_{\theta_t^{(l)}} J \circ f_{\theta_t} \right\|$$

$$+ 2\eta \sum_l \sqrt{N^{(l)}} \alpha^{(l)} \left\| m_t^{(l)} - \nabla_{\theta_t^{(l)}} J \circ f_{\theta_t} \right\| + \frac{\eta^2 \left\| NL\alpha^2 \right\|_1}{2}$$

$$(28)$$

Consequently, we can obtain the following:

$$\mathbb{E} \left[ J \circ f_{\theta_{t+1}} \right] \leq J \circ f_{\theta_t} - \eta \sqrt{N^{(l)}} \sum_l \alpha^{(l)} \left\| \nabla_{\theta_t^{(l)}} J \circ f_{\theta_t} \right\|$$

$$+ 2\eta \sqrt{N^{(l)}} \sum_l \alpha^{(l)} \mathbb{E} \left[ \left\| m_t^{(l)} - \nabla_{\theta_t^{(l)}} J \circ f_{\theta_t} \right\| \right] + \frac{N\eta^2 \left\| L\alpha^2 \right\|_1}{2}$$

$$(29)$$

Also, we can obtain the following:

$$\sum_t J \circ f_{\theta_{t+1}} \leq \sum_t J \circ f_{\theta_t} - \sum_t \eta \sum_l \sqrt{N^{(l)}} \alpha^{(l)} \mathbb{E} \left[ \left\| \nabla_{\theta_t^{(l)}} J \circ f_{\theta_t} \right\| \right]$$

$$+ \sum_t 2\eta \sum_l \sqrt{N^{(l)}} \alpha^{(l)} \mathbb{E} \left[ \left\| m_t^{(l)} - \nabla_{\theta_t^{(l)}} J \circ f_{\theta_t} \right\| \right] + \sum_t \frac{\eta^2 \left\| NL\alpha^2 \right\|_1}{2}$$

$$(30)$$

We denote the gradients of a single sample as the $\nabla \theta$ and denote the minimum and maximum values of the scaling factors $\sqrt{N^{(l)}} \alpha^{(l)}$ as the $\alpha_{min}$ and $\alpha_{min}$. We can obtain the following:

$$\frac{1}{T} \sum_{t,l} \mathbb{E} \left[ \left\| \nabla J \circ f_{\theta_t} \right\| \right]$$

$$\leq \frac{J \circ f_{\theta_0} - J \circ f_{\theta_{opt}}}{T\eta\alpha_{min}} + \frac{2\alpha_{max} \sum_{t,l} \mathbb{E} \left[ \left\| \sum_b \left( \nabla \theta_t^{(l)} - \mathbb{E} \left[ \nabla \theta_t^{(l)} \right] \right) \right\| \right]}{\alpha_{min} T b} + \frac{\eta \alpha_{max}^2 \left\| L \right\|_1}{2\alpha_{min}},$$

$$(31)$$

### A.6 GRADIENT ORTHOGONALITY

We visualize the distribution of gradient orthogonality $GO(x_i, x_j; f_\theta)$ for different data pairs $(x_i, x_j)$ in Figs. 8 and 9. The Gradient Orthogonality (GO) for model $f_\theta$ is defined as:

$$GO(x_i, x_j; f_\theta) = \frac{1}{N} \sum_l N^{(l)} \cdot Cos(|\nabla_{\theta^{(l)}} J(f_\theta(x_i))|, |\nabla_{\theta^{(l)}} J(f_\theta(x_j))|), \qquad (32)$$

where the $x_i$ and $x_j$ are two distinct samples in the subset $X$ while the $X$ denotes a subset sampled from the original dataset. The $N$ is the number of weights in the model $f_\theta$ with its weights $\theta$, while $N^{(l)}$ is the number of weights in layer $l$. The $Cos(\cdot, \cdot)$ computes cosine similarity. The $\nabla_{\theta^{(l)}} J(f_\theta(x_i))$ represents the weight gradients at layer $l$ w.r.t the loss $J(f_\theta(x_i))$.

The results in Fig. 8 suggest that the gradient orthogonality exhibits a high correlation with the model, while showing little relationship with optimizers.

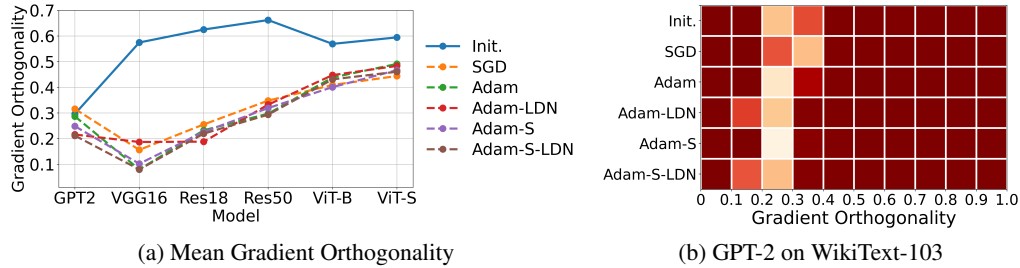

(a) Mean Gradient Orthogonality        (b) GPT-2 on WikiText-103

Figure 8: The inter-class gradient orthogonality of models trained by optimizers. **(a)**: The Mean Gradient Orthogonality (MGO) for different visual models trained on WikiText-103 or CIFAR100-LT. MGO can be calculated by Eq. 10. **(b)**: The Mean Gradient Orthogonality (MGO) for the GPT-2 trained on WikiText-103 with different optimizers.

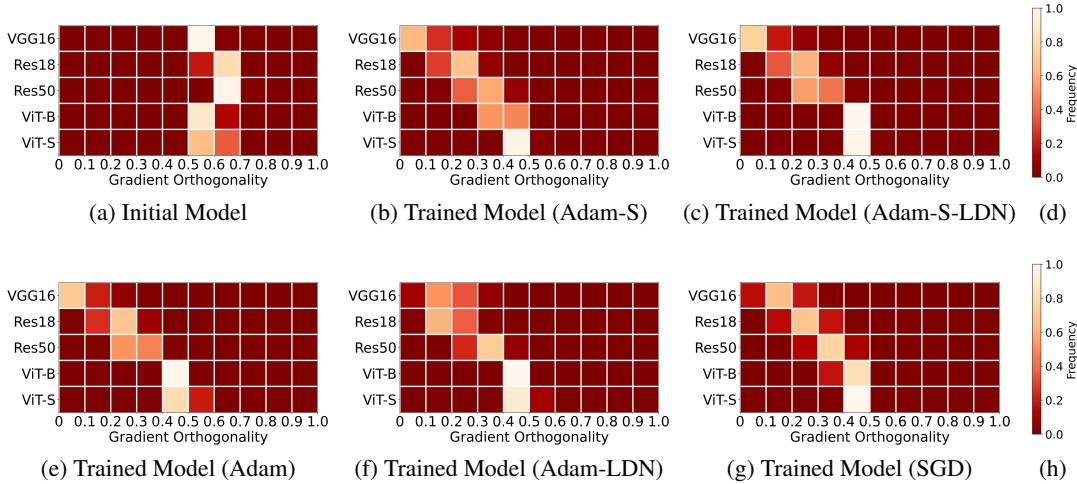

(a) Initial Model     (b) Trained Model (Adam-S)     (c) Trained Model (Adam-S-LDN)     (d)

(e) Trained Model (Adam)     (f) Trained Model (Adam-LDN)     (g) Trained Model (SGD)     (h)

Figure 9: The inter-class gradient orthogonality of models trained by optimizers on CIFAR100-LT. **(a)**: The distribution of gradient orthogonality $GO(x_i, x_j; f_\theta)$ for different data pairs $(x_i, x_j)$ at model initialization. We sample 100 data from different classes and construct 4,950 pairs. **(b)-(g)**: The distribution of gradient orthogonality for trained models. The gradient orthogonality ranges from 0 to 1, where 0 indicates full orthogonality.

## A.7 TRAINING LOSS FOR DIFFERENT IMBALANCE RATE

As shown in Figs. 10 and 11, we train models on class-imbalanced data with varying imbalance ratios to demonstrate the impact of severe data imbalance on different optimizers. The imbalance ratio is defined as the ratio of sample counts between the class with the lowest frequency and the class with the highest frequency. Specifically, the dataset used is a subset of CIFAR-100. For instance, when the imbalance ratio is set to 1/10. If the imbalance ratio is 1/20, the $c^{th}$ class contains $500 \times \frac{1}{20^\rho}$ samples. Here, the $\rho$ is uniformly sampled from the interval [0,1] with its value defined as $\rho = \frac{c}{C}$, where $C$ denotes the total number of classes.

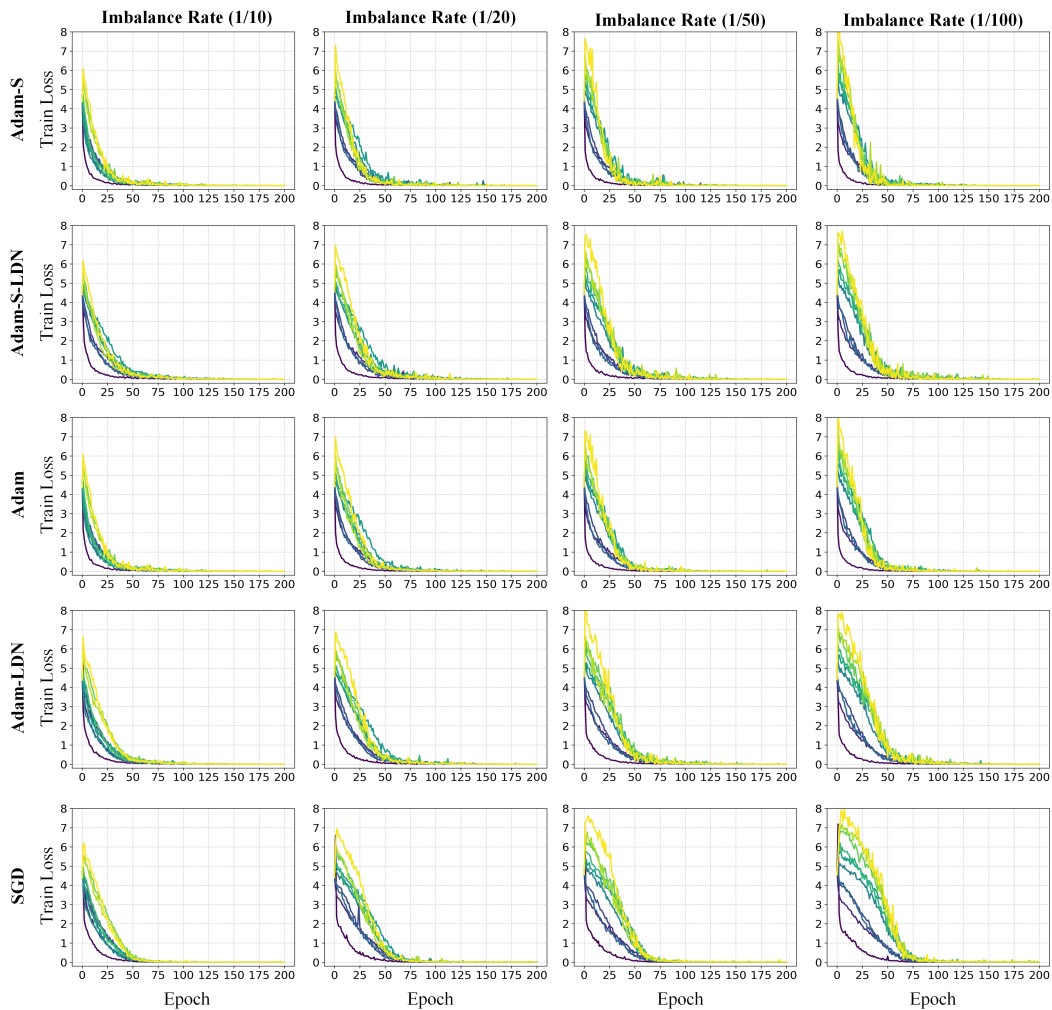

Figure 10: The training loss for 11 selected classes in a class-imbalanced dataset of different imbalance rates during model training. The trained model is ResNet18. These classes (Nos. 1, 10, 20, 30, 40, 50, 60, 70, 80, 90, 100) were sampled from a descending numerical ranking of all available classes. The imbalance ratio is defined as the ratio of sample counts between the class with the lowest frequency and the class with the highest frequency. Specifically, the dataset used is a subset of CIFAR-100. For instance, when the imbalance ratio is set to 1/10. If the imbalance ratio is 1/20, the $c^{th}$ class contains $500 \times \frac{1}{20^\rho}$ samples. Here, the $\rho$ is uniformly sampled from the interval [0,1] with its value defined as $\rho = \frac{c}{C}$, where $C$ denotes the total number of classes.

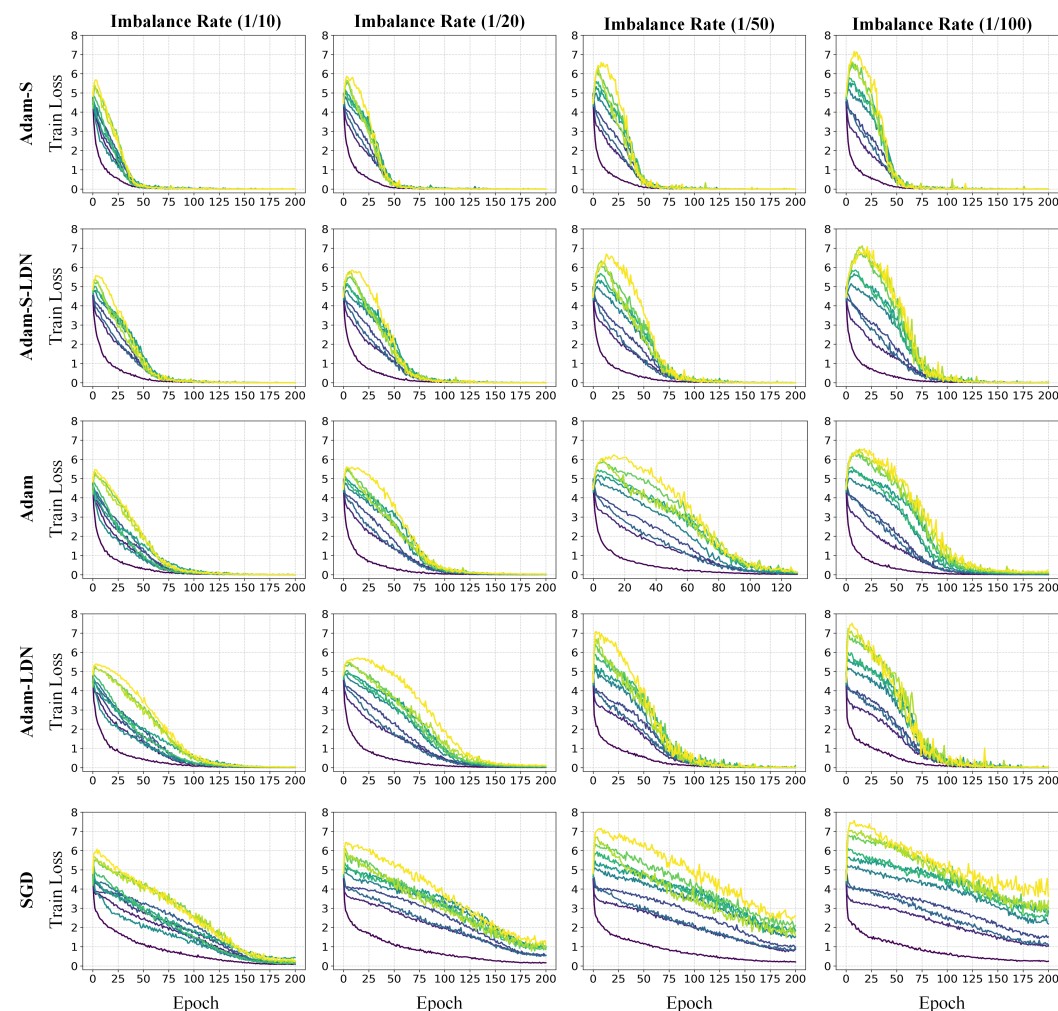

Figure 11: The training loss for 11 selected classes in a class-imbalanced dataset of different imbalance rates during model training. The trained model is ViT-S. These classes (Nos. 1, 10, 20, 30, 40, 50, 60, 70, 80, 90, 100) were sampled from a descending numerical ranking of all available classes. The imbalance ratio is defined as the ratio of sample counts between the class with the lowest frequency and the class with the highest frequency. Specifically, the dataset used is a subset of CIFAR-100. For instance, when the imbalance ratio is set to 1/10. If the imbalance ratio is 1/20, the $c^{th}$ class contains $500 \times \frac{1}{20^\rho}$ samples. Here, the $\rho$ is uniformly sampled from the interval [0,1] with its value defined as $\rho = \frac{c}{C}$, where $C$ denotes the total number of classes.

The results in Figs. 10 and 11 demonstrate that as data imbalance increases, the difficulty of model convergence rises regardless of the optimizer employed, with this effect being particularly pronounced for the SGD optimizer.

## A.8 EXPERIMENTS ON HEAVY TAILED IMAGENET

We introduced a severely class-imbalanced dataset (i.e., the heavy-tailed ImageNet) to compare optimization algorithms further, as shown in Fig. 12. Our primary focus lies in comparing the Adam-S and Adam-S-LDN optimization algorithms. The results demonstrate that both the Adam-S and Adam-S-LDN optimization algorithms exhibit similar performance. Additionally, when training on severely class-imbalanced datasets, all optimization algorithms struggle to achieve satisfactory convergence for VGG16 on the data from the low-frequency class. Compared to other models, VGG16 lacks the skip-connection, and it is perhaps this structural difference that led to poorer results. This

experiment suggests that model architectures also have a significant effect on the outcome of training class-imbalanced data.

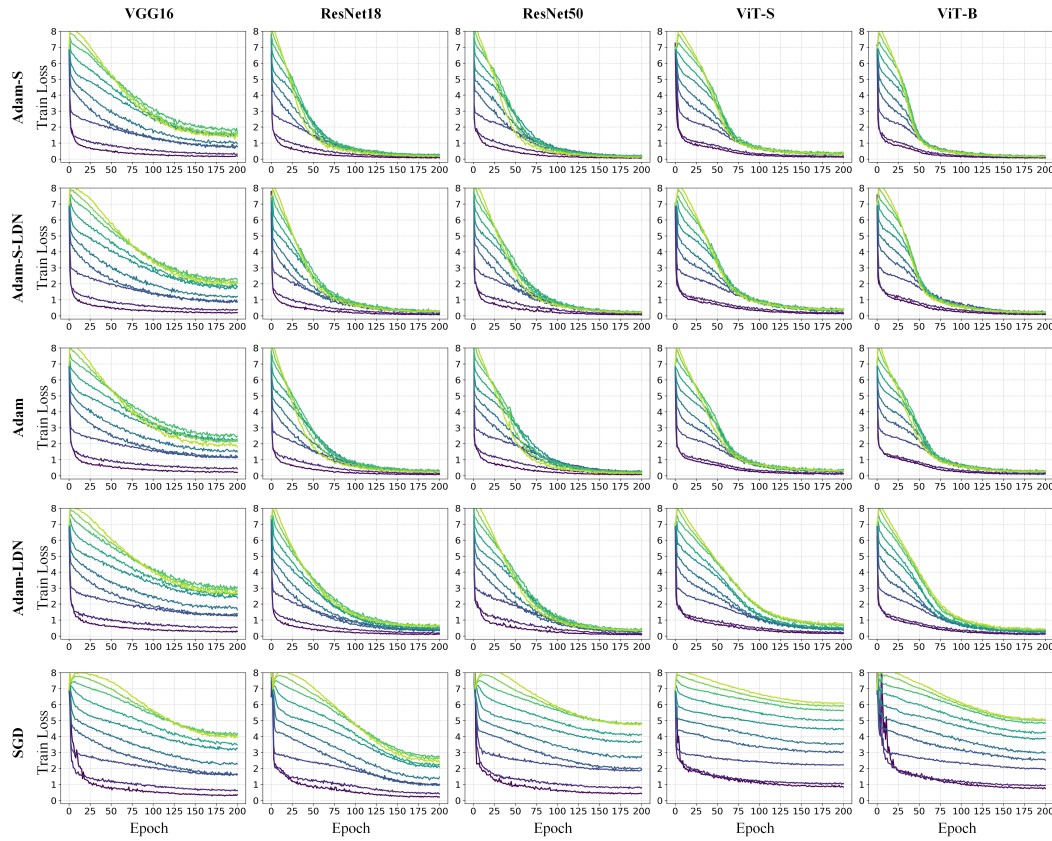

Figure 12: Comparison of Adam-S, Adam-S-LDN, Adam, Adam-LDN, and SGD when training visual models on heavy-tailed class-imbalanced data (i.e., the heavy-tailed ImageNet). We divided the data into 10 groups in order, with each group accounting for approximately 10% of the data. The groups are labelled with colors. The closer the color is to light green, the lower the frequency of the data. The heavy-tailed ImageNet is a subset of ImageNet (Russakovsky et al., 2015), subsampled to exhibit class imbalance. We sample $\left\lceil \frac{1300}{k} \right\rceil$ images from the $k^{th}$ class.

## A.9 EXPERIMENTS ON COMPUTATIONAL OVERHEAD

To demonstrate the computational overhead of different optimizers, we train GPT-2 with various optimizers on WikiText-103, as listed in Table 2. Replacing element-wise dynamics normalization with layer-level dynamics normalization in Adam yields approximately 12% savings in single-step computation time.

Table 2: Computational overhead of different optimizers. The optimizers are used to train the GPT-2 on WikiText-103.

| Optimizer | SGD | Adam | Adam-LDN | Adam-S | Adam-S-LDN |
|---|---|---|---|---|---|
| train steps per second (↑) | 2.267 | 1.683 | 1.905 | 1.653 | 1.872 |
| seconds per train step (↓) | 0.441 | 0.594 | 0.525 | 0.605 | 0.534 |
| Ratio to Adam's cost (↓) | 74.2% | 100.0% | 88.4% | 101.9% | 89.9% |

## A.10    CHANGES IN GRADIENT ORTHOGONALITY OVER TRAINING STEPS

In this section, we illustrate the change in mean gradient orthogonality over time, as shown in Fig. 13. In accordance with convention, the mean gradient orthogonality of GPT-2 varies with each step, while the mean gradient orthogonalities of ResNet18 and ViT-S vary with each epoch. The results show that during the initial stages of model training, the mean gradient orthogonality across classes usually experiences a sharp decline, subsequently remaining almost unchanged.

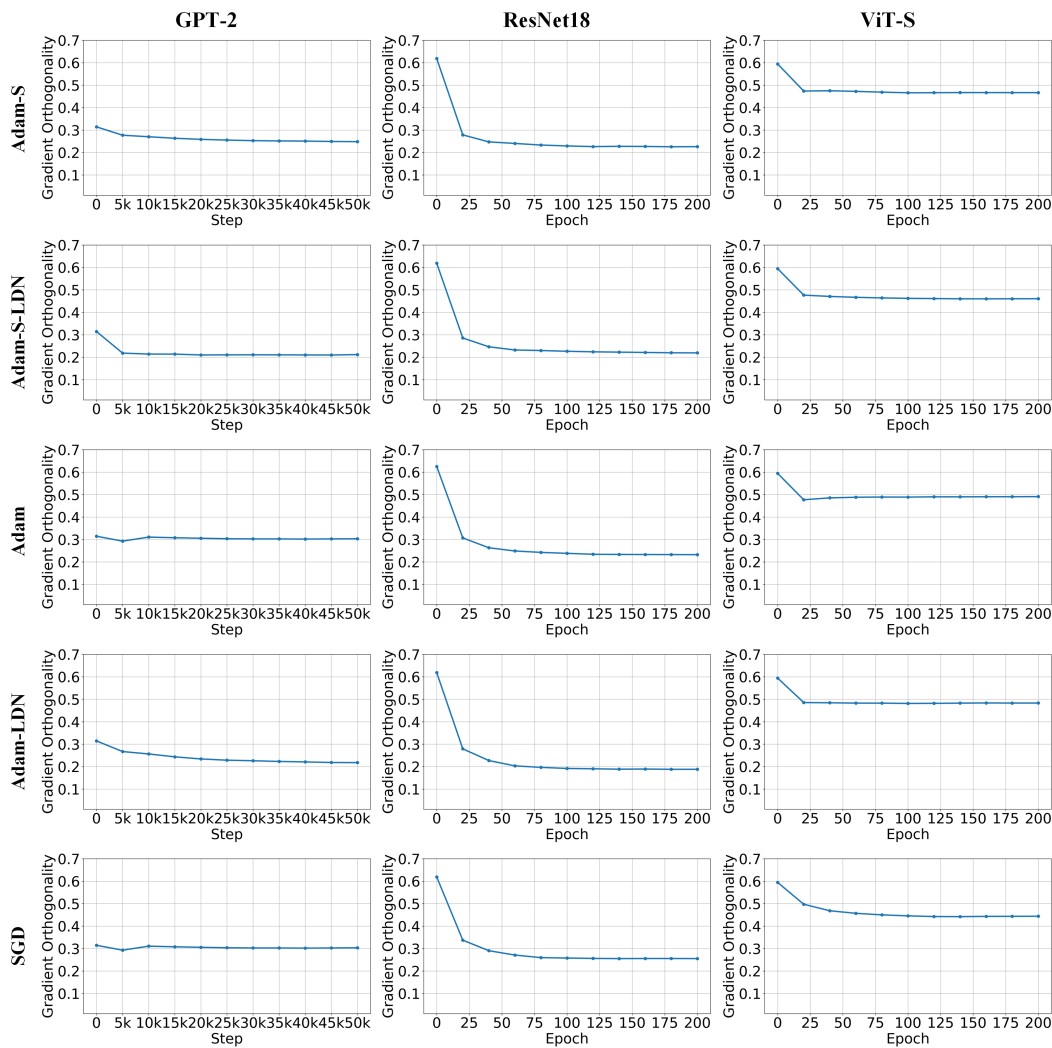

Figure 13: Change in mean gradient orthogonality over time. GPT-2 is trained on the WikiText-103 dataset, while ResNet18 and ViT-S are trained on the CIFAR100-LT dataset. In accordance with convention, the mean gradient orthogonality of GPT-2 varies with each step, while the mean gradient orthogonalities of ResNet18 and ViT-S vary with each epoch. The mean gradient orthogonality are calculated by Eq. 10.

## A.11    VISUALIZATION OF NORMALIZATION USED

For clarity, we visualize the normalization in Fig. 14.

The element-wise gradient normalization can be formulated as

$$\frac{\nabla\theta_{n,t}^{(l)}}{\sqrt{v_{n,t}^{(l)}}}, \quad v_{n,t}^{(l)} = \beta_1 \cdot v_{n,t-1}^{(l)} + (1 - \beta_1) \cdot (\nabla\theta_{n,t}^{(l)})^2, \quad v_{n,0}^{(l)} = 0, \tag{33}$$

while the layer-wise gradient normalization can be formulated as

$$\frac{\nabla\theta_{n,t}^{(l)}}{\sqrt{\frac{1}{N}\sum_n \left(\nabla\theta_{n,t}^{(l)}\right)^2}}, \tag{34}$$

where the $\nabla\theta_{n,t}^{(l)}$ is the gradients of the $n^{th}$ weight in layer $l$ at the $t^{th}$ iteration.

As shown in Fig. 14, the element-wise gradient normalization alters the form of the gradient distribution in a layer, whereas the layer-wise gradient normalization does not.

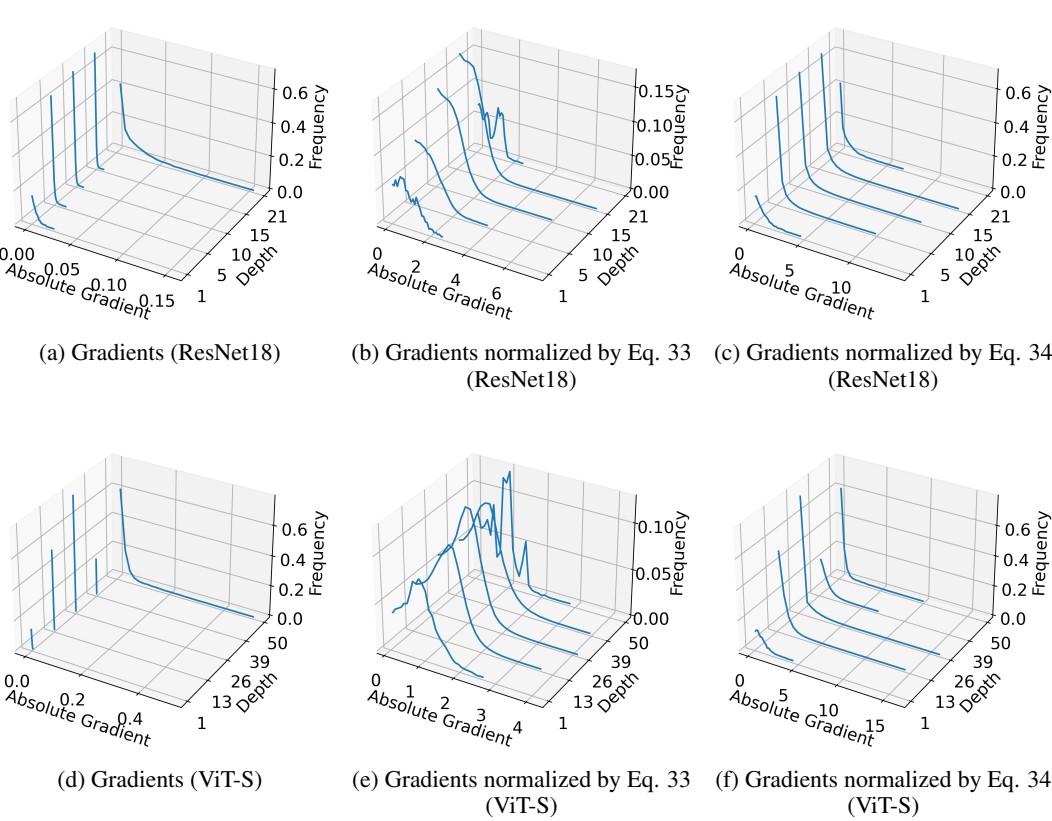

(a) Gradients (ResNet18)

(b) Gradients normalized by Eq. 33 (ResNet18)

(c) Gradients normalized by Eq. 34 (ResNet18)

(d) Gradients (ViT-S)

(e) Gradients normalized by Eq. 33 (ViT-S)

(f) Gradients normalized by Eq. 34 (ViT-S)

Figure 14: Histogram of absolute normalized gradients of weights in 5 different layers at the beginning of training. The absolute normalized gradients are calculated by element-wise gradient normalization in Eq. 33 and layer-wise gradient normalization in Eq.34, respectively.

## A.12 EXPERIMENTS ON RMSPROP

The Adam optimization algorithm is proposed by combining the momentum technique and the RMSProp optimization algorithm. RMSProp employs element-wise gradient normalisation but does not perform debiasing. To further demonstrate the role of gradient normalization under the scenario of data imbalance, we train GPT-2 with RMSProp on the WikiText-103 dataset, and train ResNet18 and ViT-S with RMSProp on the CIFAR100-LT dataset. The results are shown in Fig. 15. Models with RMSProp also fit the data from low-frequency classes well. These results suggests that dynamics

normalization can assist models coping with data imbalance while training without the momentum technique.

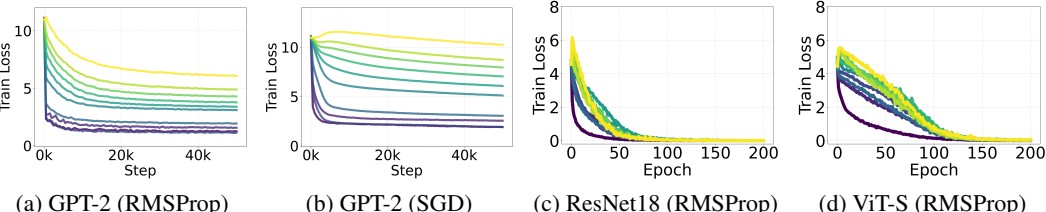

(a) GPT-2 (RMSProp)  (b) GPT-2 (SGD)  (c) ResNet18 (RMSProp)  (d) ViT-S (RMSProp)

Figure 15: Train Loss of RMSProp when training models on class-imbalanced data. The results support that dynamics normalization can assist models coping with data imbalance.

