# OpenReview forum: "Adam Can Mitigate Class Imbalance Without Element-Wise Gradient Normalization"
_ICLR.cc/2026/Conference — Submitted to ICLR 2026_

### Official Review · Reviewer_9dcu · 2025-10-31

**Soundness:** 4
**Presentation:** 4
**Contribution:** 4
**Rating:** 8
**Confidence:** 4

**Summary:**

The paper, demonstrates that Adam can mitigate class imbalance by balancing the magnitudes of gradients across iterations. It argues that the layer-wise dynamics normalization can address a layer-level in-consistency between forward propagation and back-propagation while Adam may not fully address this issue. So to tackle that the authors introduce a scaling factor proportional to the initialized weight magnitudes. In a more general sense the paper does an analysis across Adam with various theoretical and experimental results.

**Strengths:**

- Very extensive and well designed experiments in different modalities and models (both image and NLP data) with very good comparisons and variations of Adam.
- Very good presentation of the problem, the idea, and the proposal. The paper is very easy to read and digest from the audience.
- Very nice theoretical touches in the methodology of the paper that make the paper look more complete.

**Weaknesses:**

- I don't really see anything bad with the paper. I like the motivation and the analysis. The only theoretical guarantee stems in Eq. 15 if I am not mistaken right? Is there anything else that can be proven for this work? Like proof of convergence? I looked at the Appendix and couldn't find anything.

**Questions:**

What about any comparisons with RMSProp? For example in Figure 1 it's unfair to compete with SGD since we know that Adam is already an improvement. But I do see the SGD as a baseline.

In general I really enjoyed reading the paper and everything was very well organized from the abstract to the Appendix. I think this is a solid contribution to ICLR this year.

---

> ### Author Response · Authors · 2025-11-16
>
> We greatly appreciate your positive recognition of our work and valuable suggestions. We will include the convergence analysis for Adam-S-LDN and experiments on RMSProp in the Appendix.
>
> **W1: Is there anything else that can be proven for this work? Like proof of convergence?**
>
> Answer for W1: We will add the convergence analysis for Adam-S-LDN in the Appendix  (i.e., A.5 in the revised paper). Here, we present the conclusions of the analysis as follows:
>
> Assuming that the learning rate $ \eta_t =\eta $ for all { $ \eta $ }$_{t =0}^{T-1}$, the $\beta_2$ is 0 and the function $J \circ f _\theta$ is $L_l$-smooth with respect to the weights of $l$-th layer $\theta^{(l)}$, then we have the following bound:
>
> $\frac{1}{T} \sum\limits_{t,l} {\mathbb{E}\left[||\nabla J \circ f_{\theta_t} ||\right]} \leq \frac{J \circ f_{\theta_0} - J \circ f_{\theta_{opt}}}{T \eta \alpha_{min}} +\frac{2 \alpha_{max} \sum\limits_{t,l} \mathbb{E} \left[ || \sum\limits_{b} \left( \nabla {\theta_t^{(l)}} - \mathbb{E}\left[\nabla {\theta_t^{(l)}}\right] \right) || \right] }{\alpha_{min} Tb} + \frac{\eta \alpha_{max}^2 }{2 \alpha_{min}} || L ||,$
>
> where $J$ and $f_{\theta_{t}}$ represent the loss function and model with weights $\theta_{t}$. The $\theta_{opt}$ represents the optimal weights. The $\alpha_{min}$ and $\alpha_{min}$ are the minimum and maximum values of the scaling factors {${\sqrt{N^{(l)}}\alpha ^{(l)}}$}. The $b$ is the batch size. By adjusting the number of iterations $T$, batch size $b$, and learning rate $\eta$, the upper bound on the convergence rate can be reduced.
>
>
> **Q1: What about any comparisons with RMSProp? For example, similar to Fig. 1.**
>
> Answer for Q1: Thanks for your suggestion. We focus primarily on Adam due to its prevalence. Comparisons with RMSProp do indeed provide further support for our argument. Experiments similar to those depicted in Figs. 1 and 3 will be included in the appendix  (i.e., A.12 in the revised paper). The experimental results continue to support our arguments. As images cannot be displayed here, partial experimental results are provided below:
>
> Training loss for the lowest-frequency group of GPT-2 on Wikitext-103
> |  Iteration |  500 | 1k  |  2k | 5k  | 10k  |  20k | 50k  |
> |---|---|---|---|---|---|---|---|
> | SGD  | 10.972 |  10.983 | 11.054  |  11.524 |  11.497 |  11.132 | 10.250  |
> | RMSProp |  11.219 |  10.719 |  10.125 |  8.791 | 7.625  | 6.691  | 6.078  |
>
> Training loss for the lowest-frequency class of ResNet18 on Cifar100-LT
> |  Epoch |  1 | 5  |  10 | 20  | 50  |  100 | 200  |
> |---|---|---|---|---|---|---|---|
> | SGD  | 6.111 |  5.286 | 4.669  |  3.254 | 0.199  | 0.016  |  0.008 |
> | RMSProp |  6.135 | 5.166  |  3.851 |  2.267 |  0.332 | 0.046  |  0.001 |
>
> Training loss for the lowest-frequency class of ViT-S on Cifar100-LT
> |  Epoch |  1 | 5  |  10 | 20  | 50  |  100 | 200  |
> |---|---|---|---|---|---|---|---|
> | SGD  | 5.839  |  5.834 | 5.761  |  5.288 | 4.167  | 2.434  |  0.274 |
> | RMSProp |  5.367 | 5.411 |  5.171 |  4.854 |  3.389 | 0.861  |  0.089 |

---

### Official Review · Reviewer_qEga · 2025-10-31

**Soundness:** 3
**Presentation:** 3
**Contribution:** 3
**Rating:** 4
**Confidence:** 3

**Summary:**

The authors investigate why Adam reduces class imbalance problems, challenging the prior hypothesis that relies on orthogonality between gradients of different classes. They perform multiple ablations and demonstrate that Adam's success comes from normalizing gradients across iterations.

**Strengths:**

- The paper is well-written, and each section of the paper is presented with a clear focus. Their claims are well-supported, and the authors provide detailed intuitions and experiments.
- The ablations are clearly presented and well-designed. Each experiment shows evidence for / against specific hypotheses. For example, to test the importance of element-wise gradient normalization, the authors compared standard Adam with a modified version (Adam-LDN) which replaces element-wise normalization with layer-wise normalization. Because these two have very similar behaviors, this disproves the hypothesis that the benefits of Adam rely on element-wise normalization. The authors also do a good job of presenting the differences between the different variants and are explicit about what behaviors are shared.
- Adam-S-LDN has competitive performance to Adam, indicating that the authors were able to replicate the benefits of Adam into the layer-wise dynamics normalization with rescaling to account for imbalanced initialization.

**Weaknesses:**

- A significant portion of the paper is dedicated to demonstrating that gradients across classes are not orthogonal, and there are convincing experiments which illustrate this point. The author claims that these experiment demonstrate novel results which other papers disagree with. However, this seems like a strawman argument to me; my understanding of prior works is that they believe that Adam succeeds because the gradient norm and Hessian trace have strong correlation, which does not appear to be contradictory. Therefore, this decreases the novelty of this finding.
- The paper focuses on understanding Adam for class imbalance, which is a somewhat narrow field. However, their conclusions of Adam stabilizing optimization dynamics across different iterations should extend outside of this setting.

**Questions:**

- In Line 57, you claim "Kunstner et al., 2024... relies upon the assumption that the inter-class gradients are fully orthogonal". My understanding of this paper's claims was that Adam outperforms SGD in heavy-tailed class imbalance scenarios because the gradient norm and Hessian trace have high correlations, arguing that this enables Adam's normalization to behave similarly to diagonal preconditioning. Could you elaborate on why this conclusion assumes that the inter-class gradients are orthogonal?
- Previous works have found that heavy-tailed class imbalance leads to a more significant performance gap between Adam and SGD. Did you do any ablations to understand the extent of the class-imbalance on the performance of Adam vs Adam-S (and other variations)?

---

> ### Author Response · Authors · 2025-11-14
>
> **W1 & Q1: This paper claims "Kunstner et al., 2024... relies upon the assumption that the inter-class gradients are fully orthogonal". The conclusion of the paper (Kunstner et al., 2024) may not assume that the inter-class gradients are orthogonal?**
>
> Answer for W1 & Q1: First of all, we must emphasize that the work (Kunstner et al., 2024) is innovative and intriguing, offering deep insights. This is why we follow this work and try to explore it further.
>
> Sec. 3.1 in the work (Kunstner et al., 2024)  describes that "Suppose we have $c$ functions $f_1$, ..., $f_c$, corresponding to the losses for each class ...as their updates are independent of $\pi _k$ ...".  The work provides a reasonable speculation: suppose functions $f_1$, ..., $f_c$ with different weights $w_1$, ..., $w_c$ correspond to different classes $1$, ..., $c$ (which inevitably results in orthogonal gradients across classes), the updates of Adam to these functions would be independent of class frequencies. Inspired by the works (Francazi et al., 2023 and Kunstner et al., 2024), to clarify the speculation, we relax this supposition to the assumption of inter-class gradient orthogonality and generalize the quadratic problem to model optimization in general. Therefore, we consider it in fact assumes that the inter-class gradients are orthogonal.
>
> We believe that this speculation in the work (Kunstner et al., 2024) is important and valuable. If you still consider our descriptions insufficiently precise, we can adjust relevant descriptions (such as Line 57) and present the modifications here.
>
> **Q2: Did you do any ablations to understand the extent of the class-imbalance on the performance of Adam vs Adam-S (and other variations)?**
>
> Answer for Q2: You may wish to refer to the experiments in Table 1 and Fig. 7, which show the performance of Adam and Adam-S alongside SGD, Adam-LDN and Adam-S-LDN across 6 models trained on class-imbalanced data. It is worth noting that in Sec. 4.3, we demonstrate that due to layer-level inconsistencies between forward and backward propagation, gradient normalization across iterations for mitigating class imbalance must at least be a layer-wise operation. Adam-S may enhance Adam's mitigation of the inconsistencies instead of class imbalance.

---

> ### Author Response · Authors · 2025-11-27
>
> We hope the clarifications can address your concerns, and we remain committed to addressing any remaining points you may have. If you have no concerns, we kindly request you to consider updating the score.

---

### Official Review · Reviewer_DbeQ · 2025-11-01

**Soundness:** 2
**Presentation:** 2
**Contribution:** 1
**Rating:** 2
**Confidence:** 4

**Summary:**

This paper examines the factors contributing to the effectiveness of the Adam Optimizer in learning on class-imbalanced datasets. The authors demonstrate that the earlier interpretation of normalization across classes is dependent on the assumption of orthogonal gradients; therefore, the interpretation is not accurate. They show that the reason for the effectiveness is more related to the inter-iteration normalization (in the layer).

**Strengths:**

The paper is clearly readable and understandable.

**Weaknesses:**

Soundness: The authors provide plots for the training loss, but don’t give us insights into the generalization. This is problematic because the final model's usage would depend on its convergence, which is not evaluated. Hence, the effect of the proposed scaling is not observable in practice.

Formalism: The main claim of normalization across classes is only analyzed in the extreme cases of the binary classification setting, as presented in Eqs. 1 and 2, which hinders the validity of the claim. Furthermore, the claims made are not supported by sound mathematical theorems and lemmas; therefore, the validity of the claims cannot be established.

Analysis of Only the Early Phase of Training: It's unclear to me how the early phase of analysis (Fig. 3) is applicable to the final results of the model.

**Questions:**

The plot for Eq. 11 is based on Cosine Similarity for orthogonality, hence in Fig. 4a, it should be low value for orthogonality. However, the results seem to be counterintuitive. Could the authors please elaborate on that in more detail?

---

> ### Author Response · Authors · 2025-11-14
>
> **W1: Soundness: The authors provide plots for the training loss, but don't give us insights into the generalization. This is problematic because the final model's usage would depend on its convergence, which is not evaluated. Hence, the effect of the proposed scaling is not observable in practice.**
>
> Answer for W1: We show the experiments on the generalization in Table 1 and Fig. 7, which includes 6 models trained with 5 optimizers on NLP and CV tasks. Our paper investigates why Adam mitigates the class imbalance. Class imbalance can lead to optimization imbalance, which can be directly reflected in the training loss. Therefore, we mainly observe the training loss.
>
> **W2: Formalism: The main claim of normalization across classes is only analyzed in the extreme cases of the binary classification setting, as presented in Eqs. 1 and 2, which hinders the validity of the claim. Furthermore, the claims made are not supported by sound mathematical theorems and lemmas; therefore, the validity of the claims cannot be established.**
>
> Answer for W2: We analyzed GPT-2 on Wikitext103 and 5 models on CIFAR-100 and ImageNet subsets, comprising 50,257 classes, 100 classes, and 1,000 classes, respectively, rather than only focusing on binary classification. Eqs. 1 and 2 are the weight update equations with and without dynamics normalization, respectively, and have nothing to do with the question you raised.
>
> **W3: Analysis of Only the Early Phase of Training: It's unclear to me how the early phase of analysis (Fig. 3) is applicable to the final results of the model.**
>
> Answer for W3: We present the complete training process in Fig. 3, with the early phase enlarged for enhanced readability. I don't quite understand your question, but I can explain the purpose of Fig. 3. Figs. 2 and 3 are used to compare the optimization of low-frequency classes between Adam with element-wise gradient normalization and layer-wise gradient normalization (Adam vs. Adam-LDN). This motivates that Adam can mitigate class imbalance without relying on element-wise gradient normalization.
>
> **Q1: The plot for Eq. 11 is based on Cosine Similarity for orthogonality, hence in Fig. 4a, it should be low value for orthogonality. However, the results seem to be counterintuitive. Could the authors please elaborate on that in more detail?**
>
> Answer for Q1: In fact, we are demonstrating that gradients across classes are not orthogonal, rendering the assumption of inter-class orthogonality difficult to satisfy. Consequently, Adam may not rely on normalizing element-wise gradients to mitigate class imbalance. Gradient orthogonality is influenced by numerous factors, one key factor being the property of models. Specifically, CNNs that capture local high-frequency features may be more prone to exhibiting gradient orthogonality, as high-frequency features typically exhibit low similarity.

---

### Official Review · Reviewer_yQGF · 2025-11-04

**Soundness:** 3
**Presentation:** 3
**Contribution:** 2
**Rating:** 4
**Confidence:** 4

**Summary:**

This paper investigates why the Adam optimizer is able to better deal with class-imbalanced data (compared to SGD).
Previous works suggested that Adam mitigates class imbalance because its element-wise gradient normalization approximates per-class normalization, assuming gradients from different classes are orthogonal.
This paper challenges that assumption and shows empirically that inter-class gradient orthogonality is often low (especially early in training).
To study Adam, the authors introduce a variant of Adam, Adam-LDN, which removes element-wise normalization and instead performs layer-wise dynamics normalization.

Overall, I find the main argument of the paper to be convoluted. The Adam-LDN optimizer is somewhat related to ADAM but there is no proof of strong argument to explain how similar this algorithm is to ADAM so it's unclear if it is indeed a good surrogate. One part of the paper (section 4.3) also talks about layerwize normalization, but that section seems somewhat disconnected from the class imbalance problem (see questions below).

**Strengths:**

1. Challenges the gradient orthogonality assumption
2. Experiments: demonstrates results across both language (GPT-2 on WikiText-103) and vision tasks showing consistency of findings.

**Weaknesses:**

1. Limited novelty: the work primarily offers an interpretation and minor ablations of Adam rather than a substantially new optimizer
2. Lack of theoretical rigor: the analysis is largely heuristic, there is no formal convergence analysis.
3. Positioning vs related work. The proposed layer-wise scaling overlaps conceptually with prior layer-wise trust/ratio ideas and other layer-wise Adam variants (see comments/questions below).

**Questions:**

- The main argument is section 4.2 that boils down to saying that once high accuracy is achieved on 1 class, say c_1, then the relative contribution of the low-frequency class c_2 progressively dominates the weight updates. However, it seems to me that the same argument applies to normalized gradient descent if I'm not mistaken, so it's not clear to me why this explains the particular property of ADAM to mitigate class imbalance. Can you please comment on this?

- The paper claims: "To harmonize optimization dynamics, we introduce layer-specific scaling factor". However, prior works have introduced similar scaling factors I think, e.g. LAMB introduces a layer-wise trust ratio to scale updates.

- Missing prior work: the paper "Deconstructing What Makes a Good Optimizer for Language Models" by Zhao et al. (2024) proposes a variant called Adalayer which is a layer-wise variant of Adam.

- One part of the paper (section 4.3) talks about layerwize normalization, but that section seems somewhat disconnected from the class imbalance problem. Specifically, the argument there is that scaling a layer's weights by a constant factor leaves the forward output unchanged, but rescales the gradients (inversely), which creates layer-specific imbalances that can slow optimization. This is an interesting observation about optimization dynamics in general, but it seems conceptually distinct from the paper's main question: how Adam mitigates class imbalance across classes. The link to class imbalance is only loosely implied, can you elaborate on what the connection is?

Small typos:
- Equation (13) and (14): lr should be squared

---

> ### Author Response · Authors · 2025-11-16
>
> **W1: Limited novelty:  the work primarily offers an interpretation rather than a new optimizer.**
>
> Answer for W1: We do focus on mechanism interpretability, but we argue that our work is also innovative and contributive. Without understanding the mechanisms of existing optimizers, it is challenging to verify and develop an optimizer that surpasses existing optimizers (e.g., Adam), as current empirical validation struggles to cover a sufficiently broad range of scenarios. The optimization mechanism itself is a complex issue requiring long-term research, while proposing and validating a meaningful optimizer is also intricate. This may go beyond the scope of a single paper.
>
> **W2: Lack of theoretical rigor: the analysis is largely heuristic, there is no formal convergence analysis.**
>
> Answer for W2: We add the convergence analysis for Adam-S-LDN in the Appendix (i.e., A.5 in the revised paper). Here, we present the conclusions of the analysis as follows:
>
> Assuming that the learning rate $ \eta_t =\eta $ for all { $ \eta $ }$_{ t=0}^{T-1}$, the $\beta_2$ is 0 and the function $J \circ f _\theta$ is $L_l$-smooth with respect to the weights of $l$-th layer $\theta^{(l)}$, then we have the following bound:
>
> $\frac{1}{T} \sum\limits_{t,l} {\mathbb{E}\left[||\nabla J \circ f_{\theta_t}|| \right]} \leq \frac{J \circ f_{\theta_0} - J \circ f_{\theta_{opt}}}{T \eta \alpha_{min}} +\frac{2 \alpha_{max} \sum\limits_{t,l} \mathbb{E} \left[ || \sum\limits_{b} \left( \nabla {\theta_t^{(l)}} - \mathbb{E}\left[\nabla {\theta_t^{(l)}}\right] \right) || \right] }{\alpha_{min} Tb} + \frac{\eta \alpha_{max}^2 }{2 \alpha_{min}} || L ||,$
>
> where $J$ and $f_{\theta_{t}}$ represent the loss function and model with weights $\theta_{t}$. The $\theta_{opt}$ represents the optimal weights. The $\alpha_{min}$ and $\alpha_{min}$ are the minimum and maximum values of the scaling factors {$\sqrt{N^{(l)}}\alpha ^{(l)}$}. The $b$ is the batch size. By adjusting the number of iterations $T$, batch size $b$, and learning rate $\eta$, the upper bound on the convergence rate can be reduced.
>
> **Q1: The main argument in Sec. 4.2 can apply to normalized gradient descent, why this explains the particular property of ADAM to mitigate class imbalance.**
>
> Answer for Q1: We are not seeking to prove that Adam is the best optimizer. The argument in Sec. 4.2 demonstrates why Adam can mitigate class imbalance and can also apply to some normalized gradient descent methods. The argument in Sec. 4.3 demonstrates that gradient normalization must at least be a layer-wise operation in practice, otherwise it will fail to work. Replacing element-wise gradient normalization with layer-wise gradient normalization can reduce the memory expenditure of the full gradient size. We focus on interpreting Adam's mechanisms due to Adam's dominance and the readability of the paper. However, this does not imply that the conclusions are unique to Adam.
>
> **Q2 & W3: The paper claims: "To harmonize optimization dynamics, we introduce layer-specific scaling factor". Prior works have introduced similar scaling factors, e.g., LAMB.**
>
> Answer for Q2 & W3: LAMB is great work and provides a sound theoretical demonstration of the algorithm's feasibility. We describe that "the previous works (You et al. 2017; 2019) empirically identify layer-wise optimization imbalance during model training with a large batch size, and propose the LARS and LAMB optimizers to mitigate this issue." in Related Work (Lines 100-102). Sec. 4.3 demonstrates that an unfair optimization may arise from imbalanced initialization due to the inconsistencies between forward and backward propagation caused by normalization layers. Adam may not fully mitigate the unfair optimization. The introduction of a layer-level scaling factor (details in Alg. 4) proportional to the initialized weight magnitudes can mitigate this issue. To the best of our knowledge, the motivation and implementation may differ from prior works. We may further identify and explain the underlying reason behind the optimization imbalance.
>
> **Q3: Missing prior work: the paper "Deconstructing What Makes a Good Optimizer for Language Models" by Zhao et al. (2024) proposes a variant called Adalayer, which is a layer-wise variant of Adam.**
>
> Answer for Q3: Thank you for the reminder, and we cite this paper in Related Work. We insert the following sentence on Line 126: "Zhao et al. investigate the performance and stability to hyperparameters of several optimizers, suggesting that the used adaptive optimizers perform comparably."
>
> **Q4: Sec. 4.3 seems somewhat disconnected from the class imbalance. Can you elaborate on what the connection is?**
>
> Answer for Q4: Sec. 4.3 demonstrates that when employing gradient normalization to mitigate the class imbalance, normalization should be applied at least layer-wise, as normalization layers render gradient magnitudes across layers non-comparable.
>
> Thanks for the reminder regarding the typos; we added the square.

---

> ### Author Response · Authors · 2025-11-27
>
> We endeavoured to address your concerns and incorporated convergence analysis and the related work (Zhao et al. 2024) in the revised draft. If you are satisfied, we kindly request you to consider updating the score.

---

### Meta-Review · Area_Chair_TDHC · 2026-01-01

**Summary:**

This paper investigate why Adam can mitigate class imbalance. The reviewers praised the writing quality and experimental design of this paper, noting that it is well-structured with clear section focuses, well-supported arguments, and accompanied by detailed theoretical explanations and systematic ablation studies. The experiments cover multiple modalities and models (both image and NLP data) and providing thorough comparative analyses of Adam and its variants. The theoretical approach also demonstrates a certain depth, making the paper comprehensive and easy to understand.

However, the reviewers also pointed out the following limitations: the scope of the study is relatively narrow, focusing primarily on Adam's role in addressing class imbalance; the core argument is somewhat convoluted in its logical flow; additionally, although the Adam-LDN variant is proposed, there is a lack of rigorous justification regarding its similarity to the original Adam algorithm, leaving its effectiveness and theoretical basis as a substitute model unclear, which position this paper slightly below the acceptance threshold.

**Reviewer Concerns:**

Some of the reviewers' concerns have been addressed, but concerns about the scope and justification remain.

**Reviewer Scores:**

The reviewers are likely to retain their scores or slightly raise them.

---

### Decision · Program_Chairs · 2026-01-26

Reject